# The Devil is in the Wrongly-classified Samples: Towards Unified Open-set Recognition

**Jun Cen**[1*], **Di Luan**[1*], **Shiwei Zhang**[2], **Yixuan Pei**[3],
**Yingya Zhang**[2], **Deli Zhao**[2], **Shaojie Shen**[1], **Qifeng Chen**[1]
[1]Cheng Kar-Shun Robotics Institute, The Hong Kong University of Science and Technology
[2]Alibaba Group    [3]Xi'an Jiaotong University
`{jcenaa,dluan}@connect.ust.hk`, `{zhangjin.zsw,yingya.zyy,deli.zdl}@alibaba-inc.com`,
`peiyixuan@stu.xjtu.edu.cn`, `{eeshaojie,cqf}@ust.hk`

## Abstract

Open-set Recognition (OSR) aims to identify test samples whose classes are not seen during the training process. Recently, Unified Open-set Recognition (UOSR) has been proposed to reject not only unknown samples but also known but wrongly classified samples, which tends to be more practical in real-world applications. In this paper, we deeply analyze the UOSR task under different training and evaluation settings to shed light on this promising research direction. For this purpose, we first evaluate the UOSR performance of several OSR methods and show a significant finding that the UOSR performance consistently surpasses the OSR performance by a large margin for the same method. We show that the reason lies in the known but wrongly classified samples, as their uncertainty distribution is extremely close to unknown samples rather than known and correctly classified samples. Second, we analyze how the two training settings of OSR (*i.e.*, pre-training and outlier exposure) influence the UOSR. We find although they are both beneficial for distinguishing known and correctly classified samples from unknown samples, pre-training is also helpful for identifying known but wrongly classified samples while outlier exposure is not. In addition to different *training settings*, we also formulate a new *evaluation setting* for UOSR which is called few-shot UOSR, where only one or five samples per unknown class are available during evaluation to help identify unknown samples. We propose FS-KNNS for the few-shot UOSR to achieve state-of-the-art performance under all settings.

## 1 Introduction

Neural networks have achieved tremendous success in the closed-set classification (Deng et al., 2009), where the test samples share the same In-Distribution (InD) class set with training samples. Open-Set Recognition (OSR) (Scheirer et al., 2013) is proposed to tackle the challenge that some samples whose classes are not seen during training, which are Out-of-Distribution (OoD) data, may occur in the real world applications and should be rejected. However, some researchers have argued that the model should not only reject OoD samples but also InD samples that are Wrongly classified (InW), as the model gives the wrong answers for both of them. So Unified Open-set Recognition (UOSR) is proposed to only accept InD samples that are correctly classified (InC) and reject OoD and InW samples (Kim et al., 2021) simultaneously. The difference between the UOSR and OSR lies in the InW samples, where OSR is supposed to accept them while UOSR has the opposite purpose. Actually, UOSR is more useful in most real-world applications, but it receives little attention from the research community as it has been proposed very recently and lacks comprehensive systematic research. Therefore, we deeply analyze the UOSR problem in this work to fill this gap.

We first apply existing OSR methods for UOSR in Sec. 3, and then analyze UOSR under different *training settings* and *evaluation settings* in Sec. 4 and Sec. 5 respectively. In Sec. 3, several existing OSR methods are applied for UOSR, and we find that the UOSR performance is consistently and significantly better than the OSR performance for the same method, as shown in Fig. 1 (a). We show

---

*Equal contribution. Code: `https://github.com/Jun-CEN/Unified_Open_Set_Recognition`.

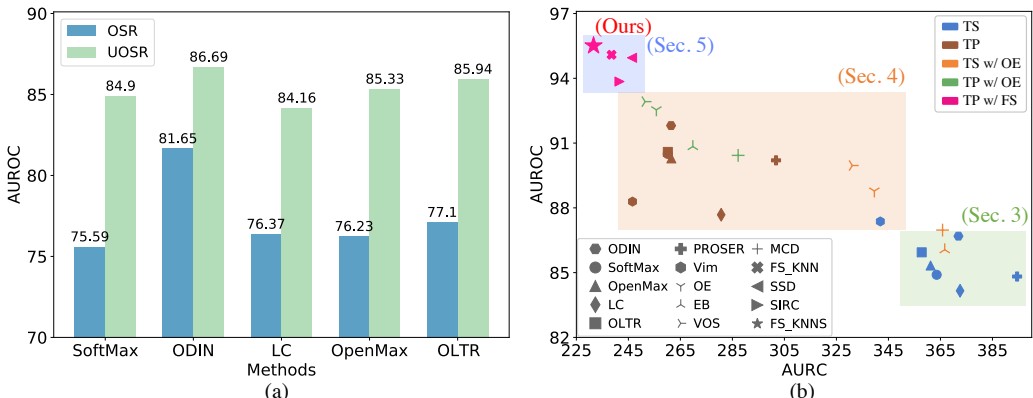

Figure 1: (a) shows that the UOSR performance is significantly better than OSR performance for the same method, which illustrates the uncertainty distribution of these OSR methods is actually closer to the expectation of UOSR than OSR. (b) shows the UOSR performance under different settings and the skeleton of this paper. Results are based on the ResNet50 backbone. CIFAR100 and TinyImageNet are InD and OoD datasets, respectively. (TS: Train from Scratch. TP: Train from Pre-training. OE: Outlier Exposure. FS: Few-shot.)

that this phenomenon holds for different network architectures, datasets, and domains (image and video recognition). We find *the devil is in the InW samples that have similar uncertainty distribution with OoD samples rather than InC samples*. Therefore, the false positive predictions in OSR tend to be InW samples, which is extremely important but dismissed by all existing OSR works.

In Sec. 4, we introduce two *training settings* into UOSR, including pre-training (Hendrycks et al., 2019a) and outlier exposure (Hendrycks et al., 2019b; Yu & Aizawa, 2019; Thulasidasan et al., 2021), as they are both helpers of the OSR that introduce extra information beyond the training set. Pre-training is to use the weights that are trained on a large-scale dataset for better down-task performance, and outlier exposure is to introduce some background data without labels into training to help the model classify InD and OoD samples. We find both of them have better performance for InC/OoD discrimination, which explains why they are beneficial for OSR. However, pre-training is also helpful for InC/InW discrimination, while outlier exposure has a comparable or even worse performance to distinguish InC and InW samples. The performance of UOSR can be regarded as the comprehensive results of InC/OoD and InC/InW discrimination so that both techniques can boost the performance of UOSR. We build up a comprehensive UOSR benchmark that involves both pre-training and outlier exposure settings, as shown in Fig. 1 (b).

In addition to the two aforementioned *training settings*, we introduce a new *evaluation setting* into UOSR in Sec. 5. We formulate the few-shot UOSR, similar to SSD (Sehwag et al., 2022) that proposes few-shot OSR, where 1 or 5 samples per OoD class are introduced for reference to better identify OoD samples. We first develop a KNN-based baseline (Sun et al., 2022) FS-KNN for the few-shot UOSR. Although InC/OoD discrimination is improved due to the introduced OoD reference samples, the InC/InW discrimination is severely harmed compared to SoftMax baseline (Hendrycks & Gimpel, 2017). To alleviate this problem, we propose FS-KNNS that dynamically fuses the FS-KNN with SoftMax uncertainty scores to keep high InC/InW and InC/OoD performance simultaneously. Our FS-KNNS achieves state-of-the-art performance under all settings in the UOSR benchmark, as shown in Fig. 1 (b), even without outlier exposure during training. Note that InC/OoD performances are comparable between FS-KNNS and FS-KNN, but their distinct InC/InW performances makes FS-KNN better at OSR and FS-KNNS better at UOSR, which illustrates the difference between few-shot OSR and UOSR and the importance of InW samples during evaluation.

## 2 TOWARDS UNIFIED OPEN-SET RECOGNITION

In this section, we first formalize the UOSR problem and then discuss the relation between UOSR and other uncertainty-related tasks.

**Unified Open-set Recognition.** Suppose the training dataset is $\mathcal{D}_{\text{train}} = \{(\mathbf{x}_i, y_i)\}_{i=1}^{N} \subset \mathcal{X} \times \mathcal{C}$, where $\mathcal{X}$ refers to the input space, *e.g.*, images or videos, and $\mathcal{C}$ refers to the InD sets. In closed-set

recognition, all test samples come from the InD sets, $i.e.$, $\mathcal{D}_{\text{test}}^{\text{closed}} = \{(\mathbf{x}_i, y_i)\}_{i=1}^{M} \subset \mathcal{X} \times \mathcal{C}$. In OSR and UOSR, the test samples may come from OoD sets $\mathcal{U}$ which are not overlap with InD sets $\mathcal{C}$, so we have $\mathcal{D}_{\text{test}}^{\text{open}} = \mathcal{D}_{\text{test}}^{\text{closed}} \cup \mathcal{D}_{\text{test}}^{\text{unknown}}$, where $\mathcal{D}_{\text{test}}^{\text{unknown}} = \{(\mathbf{x}_i, y_i)\}_{i=1}^{M'} \subset \mathcal{X} \times \mathcal{U}$. The InD test samples $\mathcal{D}_{\text{test}}^{\text{closed}}$ can be divided into two splits based on whether the sample is correctly classified or wrongly classified, $i.e.$, $\mathcal{D}_{\text{test}}^{\text{closed}} = \mathcal{D}_{\text{test}}^{\text{closed-c}} \cup \mathcal{D}_{\text{test}}^{\text{closed-w}}$, where $\mathcal{D}_{\text{test}}^{\text{closed-c}} = \{(\mathbf{x}_i, y_i) | \hat{y}_i = y_i\}_{i=1}^{M}$, $\mathcal{D}_{\text{test}}^{\text{closed-w}} = \{(\mathbf{x}_i, y_i) | \hat{y}_i \neq y_i\}_{i=1}^{M}$, and $\hat{y}_i$ refers to the model classification results of sample $\mathbf{x}_i$. The goal of UOSR is to reject InW and OoD samples and accept InC samples, so the ground truth uncertainty $u$ of $\mathcal{D}_{\text{test}}^{\text{closed-w}}$ and $\mathcal{D}_{\text{test}}^{\text{unknown}}$ is 1 while for $\mathcal{D}_{\text{test}}^{\text{closed-c}}$ it is 0, as shown in Table 1. The key of UOSR is how to estimate the uncertainty $\hat{u}$ to be close to the ground truth uncertainty $u$.

The UOSR is proposed by (Kim et al., 2021) very recently, so it has not attracted many researchers to this problem yet. SIRC (Xia & Bouganis, 2022) augments SoftMax (Hendrycks & Gimpel, 2017) baseline for the better UOSR performance. Build upon these existing methods, we make a deep analysis of UOSR in this work. In Table 1, we compare different settings of the related uncertainty estimation tasks, and the detailed discussions are as follows.

Table 1: Comparison of uncertainty-related task settings. Cls: Classification. 0 and 1 refer to the corresponding ground truth uncertainty $u$, and $u$ is not fixed in MC.

| | InC | InW | OoD | Ordinal Rank | InD Cls |
|---|---|---|---|---|---|
| SP | 0 | 1 | ✗ | ✓ | ✓ |
| AD/OD | 0 | 0 | 1 | ✓ | ✗ |
| OSR | 0 | 0 | 1 | ✓ | ✓ |
| UOSR | 0 | 1 | 1 | ✓ | ✓ |
| MC | - | - | ✗ | ✗ | ✓ |

**Selective Prediction (SP).** Apart from the classical classification, SP also tries to estimate which sample is wrongly classified (Corbière et al., 2019; Moon et al., 2020; Granese et al., 2021), so the ground truth uncertainty of InW is 1. SP is constrained under the closed-set scenario and does not consider OoD samples during evaluation, as shown in Table 1, which is the key difference with UOSR. We involve some SP methods in the UOSR benchmark in Sec. 4.

**Anomaly/Outlier Detection (AD/OD).** AD is to detect anomaly patches within an image or anomaly events within a video (Deng & Li, 2022; Zaheer et al., 2022). OD regards a whole dataset as InD and samples from other datasets as OoD (Chauhan et al., 2022; Goodge et al., 2022). Both AD and OD do not require InD classification, so there is no InC/InW discrimination problem.

**Open-set Recognition (OSR) and Out-of-distribution Detection (OoDD).** The task settings of OSR and OoDD are same, but their datasets might be different. Both of them aim to accept all InD samples no matter they are InC or InW samples, and reject OoD samples. OSR divides one dataset into two splits and uses one of them as InD data to train the model, while another split is regarded as OoD samples (Scheirer et al., 2013). In contrast, OoDD uses a whole dataset as InD data and regards another dataset as OoD data (Hendrycks & Gimpel, 2017). However, in the recent works about OSR in the video domain (Bao et al., 2021), it utilizes the OoDD setting rather than the OSR setting. In this work, we use OSR to represent the task setting and use one dataset as InD data and another dataset as OoD data. As mentioned before, the distinction between UOSR and OSR is InW samples, where OSR aims to accept them, and UOSR aims to reject them, so the ground truth uncertainty of InW samples is 0 in OSR and 1 in UOSR, as shown in Table 1. *Better InC/OoD performance is beneficial for both UOSR and OSR, but higher InC/InW discrimination is preferred by UOSR but not wanted by OSR.*

**Model Calibration (MC).** All tasks mentioned above solve the uncertainty ordinal ranking problem, $i.e.$, the ground truth uncertainty of each type of data is fixed, and the performance will be better if the estimated uncertainty is closer to the ground truth uncertainty (Geifman et al., 2019). In contrast, MC uses uncertainty to measure the probability of correctness (Guo et al., 2017). For a perfect calibrated model, if we consider all samples with a confidence of 0.8, then 80% of samples are correctly classified. In this case, 20% InW samples with a confidence of 0.8 are perfect, but in other uncertainty-related tasks, InW samples are supposed to have a confidence of either 1 or 0. MC also does not consider OoD samples. See Appendix A for more relation between UOSR and MC.

## 3 OSR Approaches for UOSR

In this section, we evaluate the existing OSR approaches for the UOSR problem, and show that the InW samples play a crucial role when evaluating the uncertainty quality. Specifically, simply changing the ground truth uncertainty $u$ of InW samples from 0 to 1 can bring a large performance

Table 2: Uncertainty distribution analysis in image domain with ResNet50. OoD dataset: Tiny-ImageNet. AUORC (%) is reported.

| Methods | InC/OoD | InC/InW | InW/OoD | OSR | UOSR |
|---------|---------|---------|---------|-----|------|
| SoftMax | 84.69 | 85.68 | 50.64 | 75.59 | 84.90 |
| ODIN | 88.35 | 80.76 | 64.36 | 81.65 | 86.69 |
| LC | 84.60 | 82.58 | 55.14 | 76.37 | 84.16 |
| OpenMax | 85.16 | 85.96 | 51.27 | 76.23 | 85.33 |
| OLTR | 85.99 | 85.74 | 52.22 | 77.10 | 85.94 |
| PROSER | 87.04 | 77.84 | 62.57 | 79.23 | 84.82 |

Table 3: Uncertainty distribution analysis in video domain with TSM backbone. OoD dataset is HMDB51. AUORC (%) is reported.

| Methods | InC/OoD | InC/InW | InW/OoD | OSR | UOSR |
|---------|---------|---------|---------|-----|------|
| OpenMax | 88.56 | 82.53 | 64.47 | 82.27 | 85.95 |
| Dropout | 85.36 | 85.86 | 48.92 | 75.75 | 85.58 |
| BNN SVI | 89.93 | 88.85 | 55.44 | 80.10 | 89.44 |
| SoftMax | 88.73 | 88.01 | 54.19 | 79.72 | 88.42 |
| RPL | 89.22 | 88.06 | 55.74 | 79.67 | 88.70 |
| DEAR | 89.33 | 85.47 | 56.78 | 80.00 | 87.56 |

Figure 2: (a) and (b) show the relation between UOSR and OSR performance in the image and video domain under ResNet50 and TSM backbones. Different color indicates different OoD datasets. The red-dotted diagonal is where UOSR has the same AUROC as OSR. Green arrows show the performance gap between UOSR and OSR for the same method.

boost, as shown in Fig. 2. Then we provide the comprehensive experiment results and discussion of this phenomenon.

**Applied Methods.** We reproduce several classical OSR methods and evaluate their UOSR and OSR performance in our experiments, including SoftMax (Hendrycks & Gimpel, 2017), ODIN (Liang et al., 2018), LC (DeVries & Taylor, 2018), OpenMax (Bendale & Boult, 2016), OLTR (Liu et al., 2019) and PROSER (Zhou et al., 2021) in the image domain, as well as DEAR (Bao et al., 2021), RPL (Chen et al., 2020), MC Dropout (Gal & Ghahramani, 2016) and BNN SVI (Krishnan et al., 2018) in the video domain.

**Datasets.** In the image domain, the InD dataset is CIFAR100 (Krizhevsky et al., 2009) and the OoD datasets are TinyImageNet (Le & Yang, 2015) and LSUN (Yu et al., 2015). In the video domain, we follow (Bao et al., 2021) and adopt UCF101 (Soomro et al., 2012) as InD dataset, while the OoD datasets are HMDB51 (Kuehne et al., 2011) and MiTv2 (Monfort et al., 2021).

**Experiments settings.** We train the network using the InD dataset and evaluate the UOSR and OSR performance based on the ground truth in Table 1. The evaluation metric is AUORC (Hendrycks & Gimpel, 2017) which is a threshold-free value. The AUROC reflects the distinction quality of two uncertainty distributions. We adopt VGG13 (Simonyan & Zisserman, 2015) and ResNet50 (He et al., 2016) as the network backbone in the image domain and TSM (Lin et al., 2019) and I3D (Carreira & Zisserman, 2017) in the video domain. More experiment settings are in Appendix E.

**Results.** We provide the UOSR and OSR performance of different methods in Fig. 2. It shows that all data points are above the red dotted diagonal, illustrating that the UOSR performance is significantly better than the OSR performance of the same method, and this relationship holds across different datasets, domains (image and video), and network architectures (See Appendix C for results of more network backbones). The performance gap between UOSR and OSR of the same method can be very large, such as 8.84% for OLTR when TinyImageNet is the OoD dataset, and 10.16% for BNN SVI when the OoD dataset is MiTv2. Therefore, existing OSR methods have uncertainty distributions that are actually closer to the expectation of UOSR than OSR.

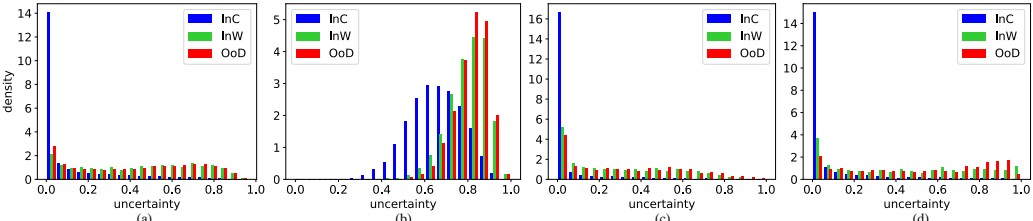

Figure 3: (a) and (b) are the SoftMax and ODIN methods in the image domain, while (c) and (d) are the SoftMax and DEAR methods in the video domain. OoD datasets are TinyImageNet for the image domain and HMDB51 for the video domain.

**Analysis.** To better understand our findings, we provide a detailed analysis of the uncertainty distribution relationships between InC, InW, and OoD samples in Tables 2 and 3. Note that higher AUROC means a better distinction between two uncertainty distributions, and AUROC=50% means two distributions overlap with each other. From Tables 2 and 3 we can clearly see that AUROC of InC/OoD and InC/InW are significantly higher than InW/OoD, and AUROC of InW/OoD is very close to 50%. Therefore, the uncertainty distribution of InC samples is distinguishable from OoD and InW samples, and there is a lot of overlap between the uncertainty distributions of InW and OoD samples. Several uncertainty distribution visualizations are in Fig. 3, where we can see that InW and OoD samples share a similar uncertainty distribution, while InC samples have smaller uncertainty.

**Importance.** Based on the above analysis, we conclude that the false positive predictions in OSR tend to be InW samples, since they share similar uncertainty distributions with OoD samples. This conclusion is extremely important as InW samples significantly deteriorate the OSR performance. Without InW samples, the OSR performance increases by a large margin (75.59 to 84.69 for Soft-Max method in Table 2). However, no existing OSR works mentioned and considered this phenomenon. We explicitly point out this conclusion and hope the following researchers take it into account when they design new OSR or UOSR methods.

**Why?** To deeply understand why InW samples share similar uncertainty distribution with OoD samples instead of InC samples, we begin by analyzing the feature distributions of InC/InW/OoD samples. We first find that the features of InC/InW/OoD follow the hierarchy structure where InC/InW/OoD samples are gradually far away from the training samples, so their features are separable. Please see Fig. 10 and 11 for reference. Then we calculate the feature similarity and surprisingly find that InW features are more similar to InC features rather than OoD features in Table 9, which contradicts the uncertainty score phenomenon. Therefore, the reason that InW samples have similar uncertainty scores with OoD samples is not they have similar features, but lies in the uncertainty estimation methods. We provide explanations about how the uncertainty estimation process causes the contradictory phenomenon in Appendix D.

**A better closed-set model is better for UOSR.** Recently, (Vaze et al., 2022) found that better closed-set performance means better OSR performance. We provide a deeper explanation of this finding and show that this conclusion also holds for UOSR. Table 4 shows that we improve the closed-set accuracy through data augmentation and longer training (Vaze et al., 2022). The

Table 4: Relation between closed-set accuracy *Acc.* (%) and open-set performance. *Aug:* Augmentation; *Ep:* Epoch. AUROC (%) is reported.

| Aug. | Ep. | Acc. | InC/OoD | InC/InW | OSR | UOSR |
|---|---|---|---|---|---|---|
| ✗ | 100 | 59.41 | 81.65 | 82.47 | 68.64 | 81.89 |
| ✓ | 100 | 68.83 | 84.18 | 85.13 | 73.71 | 84.40 |
| ✓ | 300 | 73.28 | 84.69 | 85.68 | 75.59 | 84.90 |

open-set method is the SoftMax baseline with ResNet50 backbone, and the OoD dataset is Tiny-ImageNet. For OSR, we can see that AUROC of InC/OoD is significantly better than OSR, which indicates the uncertainty distribution of InW samples are contradictory with the expectation of OSR. So less InW samples and better InC/OoD performance are two reasons for better OSR performance when closed-set accuracy is higher. For UOSR, both the InC/InW and InC/OoD performance are improving with the growth of closed-set accuracy, which brings better UOSR performance.

## 4  PRE-TRAINING AND OUTLIER EXPOSURE

After directly applying existing OSR methods for UOSR in Sec. 3, we explore two additional training settings in this section, including pre-training (Hendrycks et al., 2019a) and outlier ex-

Table 5: UOSR benchmark in the image domain under the ResNet50 model. InD dataset is CI-FAR100 while the OoD dataset is TinyImageNet. $\dagger, \ddagger, \Diamond$ refer to OSR-based, SP-based, UOSR-based methods. OD: Outlier Data. N/G/R means No/Generated/Real OD. AUORC (%), AURC ($\times 10^3$) and Acc. (%) are reported.

| Methods | OD | w/o pre-training | | | | w/ pre-training | | | |
|---|---|---|---|---|---|---|---|---|---|
| | | UOSR | | | OSR | UOSR | | | OSR |
| | | Acc.↑ | AURC↓ | AUROC↑ | AUROC↑ | Acc.↑ | AURC↓ | AUROC↑ | AUROC↑ |
| ODIN[†] | N | 72.08 | 371.92 | 86.69 | 81.65 | 86.48 | 261.42 | 91.81 | 91.47 |
| LC[†] | N | 72.08 | 372.55 | 84.16 | 76.37 | 86.48 | 280.69 | 87.68 | 85.16 |
| OpenMax[†] | N | 73.64 | 361.22 | 85.33 | 76.23 | 86.43 | 261.53 | 90.29 | 85.86 |
| MaxLogits[†] | N | 73.89 | 351.39 | 87.09 | 80.11 | 86.42 | 249.86 | 92.91 | 91.11 |
| Entropy[†] | N | 73.89 | 355.54 | 86.25 | 78.14 | 86.43 | 257.36 | 91.21 | 87.22 |
| OLTR[†] | N | 73.69 | 357.79 | 85.94 | 77.10 | 86.22 | 260.17 | 90.59 | 86.05 |
| Ensemble[†] | N | 76.78 | 314.56 | 89.59 | 83.93 | 87.41 | 240.34 | 93.67 | 92.02 |
| Vim[†] | N | 73.89 | 341.88 | 87.37 | 80.92 | 86.43 | 246.59 | 93.01 | 92.33 |
| BCE[‡] | N | 73.29 | 369.91 | 84.29 | 74.59 | 86.39 | 257.74 | 90.79 | 86.50 |
| TCP[‡] | N | 71.80 | 369.17 | 85.17 | 75.46 | 86.83 | 261.97 | 89.95 | 85.67 |
| DOCTOR[‡] | N | 72.08 | 378.45 | 84.48 | 75.06 | 86.48 | 262.05 | 90.24 | 85.75 |
| SIRC(MSP, $\|z\|_1$)[◊] | N | 73.13 | 358.93 | 85.77 | 76.67 | 86.44 | 260.08 | 90.53 | 85.98 |
| SIRC(MSP, Res.)[◊] | N | 73.13 | 348.40 | 87.26 | 80.29 | 86.44 | 247.75 | 92.99 | 90.39 |
| SIRC($-\mathcal{H}$, $\|z\|_1$)[◊] | N | 73.13 | 355.62 | 86.57 | 78.16 | 86.44 | 257.28 | 91.23 | 87.27 |
| SIRC($-\mathcal{H}$, Res.)[◊] | N | 73.13 | 346.21 | 87.71 | 81.43 | 86.44 | 244.55 | 93.68 | 91.62 |
| SoftMax[†] | N | 73.28 | 363.55 | 84.90 | 75.59 | 86.44 | 260.14 | 90.50 | 85.93 |
| SoftMax[†](OE[†]) | R | 73.54 | 339.59 | 88.78 | 84.35 | 85.43 | 255.77 | 92.54 | 90.65 |
| ARPL[†] | N | 73.03 | 345.84 | 88.13 | 80.49 | 84.67 | 301.27 | 87.01 | 83.49 |
| ARPL+CS[†] | R | 72.78 | 349.50 | 87.65 | 82.81 | 83.60 | 268.00 | 92.00 | 91.17 |
| MCD[†](Dropout[†]) | N | 76.49 | 375.01 | 82.25 | 79.21 | 87.21 | 301.71 | 83.57 | 81.69 |
| MCD[†] | R | 70.88 | 365.82 | 86.97 | 79.88 | 81.96 | 287.21 | 90.43 | 86.59 |
| PROSER[†] | G | 68.08 | 394.48 | 84.82 | 79.23 | 81.32 | 301.78 | 90.20 | 89.29 |
| PROSER[†](EB[†]) | R | 71.82 | 366.65 | 86.06 | 81.95 | 85.06 | 269.79 | 90.84 | 90.38 |
| VOS[†] | G | 73.44 | 356.68 | 86.65 | 79.72 | 86.62 | 249.24 | 92.94 | 91.37 |
| VOS[†] | R | 73.18 | 331.12 | 89.96 | 85.78 | 85.93 | 251.44 | 92.92 | 91.34 |
| OpenGAN[†] | R | 73.61 | 334.04 | 88.92 | 85.48 | 86.25 | 260.19 | 91.21 | 90.94 |

posure (Hendrycks et al., 2019b), which are effective methods to improve the OSR performance because of introduced extra information beyond the training set. Pre-training is to use large-scale pre-trained weights for initialization for better down-task performance. Outlier exposure is to introduce some unlabeled outlier data (OD) during training and regard these outlier data as the proxy as OoD data to improve the open-set performance. We find both of them also have a positive effect on UOSR, but for different reasons.

**Pre-training settings.** In the image domain, we use BiT pre-training weights (Kolesnikov et al., 2020) for ResNet50 and ImageNet (Deng et al., 2009) pre-training weights for VGG13. In the video domain, Kinetics400 (Carreira & Zisserman, 2017) pre-trained weights are used for initialization.

**Outlier exposure settings.** In the image domain, we use 300K Random Images dataset (Hendrycks et al., 2019b) as outlier data. In the video domain, outlier data comes from Kinetics400, and we filter out the overlapping classes with InD and OoD datasets. We implement several outlier exposure based methods, including OE (Hendrycks et al., 2019b), EB (Thulasidasan et al., 2021), VOS (Du et al., 2022) and MCD (Yu & Aizawa, 2019).

**UOSR benchmark settings.** We include OSR-based, SP-based, and UOSR-based methods to build a comprehensive UOSR benchmark. We implement several Selective Prediction (SP) based methods, including BCE, TCP (Corbière et al., 2019), CRL (Moon et al., 2020), and DOCTOR (Granese et al., 2021). Although these SP methods are originally designed to differentiate InC and InW samples, we adopt them in the UOSR setting to build a more comprehensive benchmark. The only UOSR-based method is SIRC (Xia & Bouganis, 2022), which combines two uncertainty scores for better UOSR performance. The evaluation metrics are AURC and AUROC (Kim et al., 2021). More details can be found in Appendix E.

Table 6: UOSR benchmark in the video domain under the TSM model. InD dataset is UCF101 while the OoD dataset is HMDB51. $\dagger, \ddagger, \diamond$ refer to OSR-based, SP-based, UOSR-based methods. OD: Outlier Data. AUORC (%), AURC ($\times 10^3$) and Acc. (%) are reported.

| | | w/o pre-training | | | | w/ pre-training | | | |
| | | UOSR | | | OSR | UOSR | | | OSR |
| Methods | OD | Acc.↑ | AURC↓ | AUROC↑ | AUROC↑ | Acc.↑ | AURC↓ | AUROC↑ | AUROC↑ |
|---|---|---|---|---|---|---|---|---|---|
| OpenMax$^\dagger$ | N | 73.92 | 185.81 | 85.95 | 82.27 | 95.32 | 75.75 | 91.22 | 90.89 |
| BNN SVI$^\dagger$ | N | 71.51 | 181.45 | 89.44 | 80.10 | 94.71 | 69.89 | 93.58 | 91.81 |
| RPL$^\dagger$ | N | 71.46 | 186.18 | 88.70 | 79.67 | 95.59 | 72.88 | 92.44 | 90.53 |
| DEAR$^\dagger$ | N | 71.33 | 215.80 | 87.56 | 80.00 | 94.41 | 102.01 | 91.50 | 91.49 |
| BCE$^\ddagger$ | N | 69.90 | 223.57 | 83.27 | 78.96 | 93.66 | 110.42 | 83.83 | 81.64 |
| CRL$^\ddagger$ | N | 71.80 | 183.76 | 88.75 | 78.57 | 95.22 | 67.61 | 93.36 | 91.38 |
| DOCTOR$^\ddagger$ | N | 72.01 | 182.04 | 88.73 | 79.76 | 95.06 | 65.61 | 93.89 | 91.80 |
| SIRC(MSP, $\|z\|_1$)$^\diamond$ | N | 73.59 | 173.32 | 88.71 | 80.33 | 95.00 | 65.97 | 93.74 | 91.42 |
| SIRC(MSP, Res.)$^\diamond$ | N | 73.59 | 174.76 | 88.17 | 78.74 | 95.00 | 65.43 | 93.83 | 91.73 |
| SIRC($-\mathcal{H}$, $\|z\|_1$)$^\diamond$ | N | 73.59 | 172.30 | 89.27 | 81.61 | 95.00 | 66.11 | 94.06 | 91.95 |
| SIRC($-\mathcal{H}$, Res.)$^\diamond$ | N | 73.59 | 175.79 | 88.50 | 79.62 | 95.00 | 69.06 | 94.07 | 92.18 |
| SoftMax$^\dagger$ | N | 73.92 | 173.14 | 88.42 | 79.72 | 95.03 | 68.08 | 93.94 | 91.75 |
| SoftMax$^\dagger$(OE$^\dagger$) | R | 74.42 | 162.36 | 90.19 | 86.29 | 94.71 | 67.93 | 94.33 | 93.40 |
| MCD$^\dagger$(Dropout$^\dagger$) | N | 73.63 | 184.66 | 85.58 | 75.75 | 95.06 | 79.53 | 90.30 | 88.23 |
| MCD$^\dagger$ | R | 72.49 | 168.83 | 91.26 | 85.57 | 93.47 | 71.19 | 95.34 | 93.68 |
| VOS$^\dagger$ | G | 74.00 | 187.82 | 86.10 | 84.51 | 95.27 | 65.68 | 94.44 | 93.62 |
| VOS$^\dagger$ | R | 74.68 | 172.71 | 87.98 | 87.09 | 94.79 | 64.99 | 94.97 | 93.72 |
| EB$^\dagger$ | R | 70.90 | 212.01 | 85.32 | 86.47 | 94.66 | 67.83 | 94.40 | 93.06 |

Figure 4: (a) and (b) plot the InC/InW and InC/OoD discrimination in the image and video domain. We set the SoftMax method training from scratch as the original point and divide the coordinate system into 4 quadrants (Q1 to Q4). (TS: Train from Scratch. TP: Train from Pre-training. OE: Outlier Exposure.)

**Results.** The UOSR benchmarks for image and video domains are Table 5 and 6 respectively. See Appendix J for the results of different model architectures. First, we can see the AUROC of UOSR is higher than OSR for almost all the methods under both settings, *i.e.*, pre-training and outlier exposure, which further strengthens the conclusion that the uncertainty distribution of OSR methods is closer to the ground truth of UOSR. For instance, AUROC is 89.59 for UOSR and 83.93 for OSR under the Ensemble method w/o pre-training in Table 5. Second, pre-training and outlier exposure can effectively boost the UOSR and OSR performance , *e.g.*, the pre-training boost the AUROC of UOSR/OSR from 84.90/75.59 to 90.50/85.93 for the SoftMax, and outlier exposure method OE has 90.19/86.29 AUROC of UOSR/OSR compared to 84.90/75.59 of SoftMax in Table 5. Note that to ensure outlier data is useful, we keep methods w/ and w/o outlier data as similar as possible, like SoftMax/OE, ARPL/ARPL+CS, and Dropout/MCD. Third, real outlier data is more beneficial than generated outlier data in the UOSR and OSR tasks. The UOSR/OSR AUROC of VOS method is 89.96/85.78 for real outlier data and 86.65/79.72 for generated outlier data, provided in Table 5.

**Analysis.** To better understand why pre-training and outlier exposure are helpful for UOSR and OSR, we provide the InC/InW and InC/OoD discrimination performance of each method in Fig. 4.

Table 7: Results of few-shot UOSR in the image domain. Model is ResNet50 with pre-training. InD and OoD datasets are CIFAR100 and TinyImageNet. AUORC (%) and AURC ($\times 10^3$) are reported.

| Methods | 5-shot AURC↓ UOSR | 5-shot AUROC↑ UOSR | OSR | InC/OoD | InC/InW | 1-shot AURC↓ UOSR | 1-shot AUROC↑ UOSR | OSR | InC/OoD | InC/InW |
|---|---|---|---|---|---|---|---|---|---|---|
| SoftMax | 260.14 | 90.51 | 85.93 | 90.64 | 89.58 | 260.14 | 90.51 | 85.93 | 90.64 | 89.58 |
| KNN | 245.48 | 93.45 | 92.34 | 94.86 | 83.08 | 245.48 | 93.45 | 92.34 | 94.86 | 83.08 |
| FS-KNN | 238.54 | 95.09 | 95.91 | 97.20 | 79.58 | 239.71 | 94.67 | 94.91 | 96.52 | 81.04 |
| SSD | 246.42 | 94.95 | **97.89** | **98.32** | 70.14 | 245.99 | 94.16 | **96.51** | **97.16** | 72.09 |
| FS-KNN+S | 239.49 | 94.14 | 91.64 | 95.27 | 85.85 | 240.30 | 93.97 | 91.36 | 94.99 | 86.45 |
| FS-KNN*S | 255.82 | 91.26 | 87.10 | 91.47 | **89.67** | 255.69 | 91.30 | 87.16 | 91.51 | **89.69** |
| SIRC | 241.58 | 93.85 | 91.42 | 94.44 | 89.58 | 239.64 | 94.20 | 92.03 | 94.87 | 89.26 |
| FS-KNNS | **231.61** | **95.51** | 94.16 | 96.54 | 87.98 | **234.84** | **94.91** | 93.10 | 95.84 | 88.08 |

We can see that almost all outlier exposure methods are in the Q2, which means that outlier exposure methods have lower InC/InW AUROC and higher InC/OoD AUROC than the SoftMax baseline. For OSR, both lower InC/InW and higher InC/OoD AUROC are beneficial, but only higher InC/OoD AUROC is wanted for UOSR, while lower InC/InW AUROC is not. This can explain why some of the UOSR performances with outlier exposure are comparable or even worse than the baseline, such as EB and VOS in the video domain Table 6, but all of the OSR performances are increased. In contrast, pre-training is helpful for both InC/InW and InC/OoD AUROC, as methods with pre-training are in the Q1. Outlier exposure methods also benefit from pre-training, as green marks are at the upper right location compared to orange marks. Therefore, outlier exposure and pre-training bring the UOSR performance gain for different reasons. InC/OoD performance is improved by both of the techniques, but pre-training is also helpful for InC/InW, while outlier exposure is not. This can be explained by the fact that the model may see additional InD samples during pre-training so that the closed-set accuracy and InC/InW discrimination are improved. In contrast, outlier exposure only provides OoD data but no more InD data, so only InC/OoD performance is improved.

## 5 FEW-SHOT UNIFIED OPEN-SET RECOGNITION

In addition to the analysis of two useful training settings in Sec. 4, we introduce a new evaluation setting into UOSR in this section. Inspired by the recent work SSD (Sehwag et al., 2022) which proposes the few-shot OSR, we introduce the few-shot UOSR, where 1 or 5 samples per OoD class can be introduced for reference to better distinguish OoD samples. The introduced reference OoD datasets are marked as $\mathcal{D}_{test}^{ref} = \{(\mathbf{x}_i, y_i)\}_{i=1}^{N'} \subset \mathcal{X} \times \mathcal{U}$. We show that few-shot UOSR has distinct challenges from few-shot OSR because of InW samples, and we propose our FS-KNNS method to achieves the state-of-the-art UOSR performance even without outlier exposure during training. **Baselines and dataset settings.** SSD is the only existing few-shot OSR method that utilizes the Mahalanobis distance with InD training samples $\mathcal{D}_{train}$ and OoD reference samples $\mathcal{D}_{test}^{ref}$. KNN (Sun et al., 2022) utilizes the feature distance with samples in $\mathcal{D}_{train}$ as uncertainty scores and shows it is better than Mahalanobis distance. So we slightly modify KNN to adapt to the few-shot UOSR and find the modified FS-KNN can already beat SSD. $\mathcal{D}_{test}^{ref}$ comes from the validation set of TinyImageNet and training set of UCF101 in the image and video domain. We repeat the evaluation process until all OoD reference samples are used and calculate the mean as the final few-shot result.

**FS-KNN.** Given a test sample $\mathbf{x}^*$ and a feature extractor $f$, the feature of $\mathbf{x}^*$ is $\mathbf{z}^* = f(\mathbf{x}^*)$. The cosine similarity set between $\mathbf{z}^*$ and the feature set of $\mathcal{D}_{train}$ is $\mathcal{S}_{train} = \{d_i\}_{i=1}^{N}, d_i = (\mathbf{z}^* \cdot \mathbf{z}_i)/(\|\mathbf{z}^*\| \cdot \|\mathbf{z}_i\|), \mathbf{z}_i = f(\mathbf{x}_i), \mathbf{x}_i \in \mathcal{D}_{train}$. $\mathcal{S}_{test}^{ref}$ is similar with $\mathcal{S}_{train}$ except $\mathbf{x}_i$ comes from $\mathcal{D}_{test}^{ref}$. We believe the uncertainty should be higher if the test sample is more similar to reference OoD samples and not similar with InD samples. Therefore, the uncertainty score is

$$\hat{u}_{fs-knn} = 1 - topK(S_{train}) + topK(S_{test}^{ref}), \tag{1}$$

where $topK$ means the $K^{th}$ largest value. The performances of SoftMax, KNN, FS-KNN, and SSD are in Table 7 and 19. Pre-training is used for better performance. FS-KNN has better InC/OoD performance than KNN as reference OoD samples are introduced. Although the overall UOSR performance of FS-KNN is better than the SoftMax baseline, *the InC/InW performance is significantly*

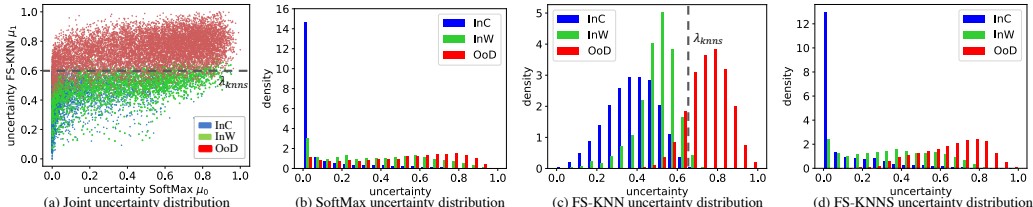

Figure 5: Uncertainty scores of each test sample (a) and uncertainty distribution of SoftMax (b), FS-KNN (c), and FS-KNNS (d).

*sacrificed, which is also an important aspect of UOSR.* For example, InC/InW performance drops from 89.58 to 79.58 in the 5-shot results of Table 7. So we naturally ask a question: *Can we improve the InC/OoD performance based on the introduced OoD reference samples while keeping similar InC/InW performance with SoftMax baseline?* This is the key difference between few-shot OSR and few-shot UOSR, as low InC/InW performance is wanted in OSR but not preferred in UOSR.

**FS-KNNS.** Inspired by SIRC (Xia & Bouganis, 2022), we aim to find a way to fuse SoftMax and FS-KNN scores so that the mixed score can keep the high InC/OoD performance of FS-KNN and meanwhile has the comparable InC/InW performance with SoftMax. The uncertainty distributions of SoftMax and FS-KNNS are depicted in Fig. 5. We find that OoD samples have larger uncertainty than InW and InC samples in the FS-KNN method, but the uncertainty of InC samples overlaps a lot with InW samples, which explains the reason that InC/OoD performance is high but InC/InW performance is low. In contrast, InW and OoD samples share similar uncertainty in the SoftMax method, which brings higher InC/InW performance. Therefore, we want to keep the uncertainty of InW samples in the SoftMax method, as well as the uncertainty of OoD samples in the FS-KNN method. In this way, the mixed scores obtain the high InC/OoD performance from FS-KNN while keeping the comparable InC/InW performance of SoftMax. We call this method FS-KNN with SoftMax (FS-KNNS), and the uncertainty is

$$\hat{u}_{fs-knns} = u_0 + \frac{1}{1 + e^{-\alpha(u_1 - \lambda_{knns})}} u_1, \tag{2}$$

where $u_0$ and $u_1$ refer to the uncertainty score of SoftMax and FS-KNN, respectively. $\lambda_{knns}$ is a threshold to determine when the weight of $u_1$ becomes large, and $\alpha$ is a coefficient to control the change rate of the weight. $\hat{u}_{fs-knns}$ will be largely influenced by $u_1$ when $u_1 > \lambda_{knns}$. A proper $\lambda_{knns}$ should be located between the InW and OoD samples, as shown in Fig. 5 (a) and (c). In this way, the uncertainty of InW samples is mainly controlled by SoftMax, and OoD samples are strengthened by FS-KNN. Ablation study about hyperparameters $K, \alpha$, and $\lambda_{knns}$ is in Appendix L.

**Discussion.** The uncertainty distribution of FS-KNNS is shown in Fig. 5 (d), which shows the uncertainty of InC and InW samples are similar to SoftMax (b), but the uncertainty of OoD samples is larger. From Table 7 and 19, we can see our FS-KNNS has significantly better InC/InW performance than FS-KNN (87.98 and 79.58 under ResNet50), and meanwhile keeps the high InC/OoD performance, so the overall UOSR performance is better than both SoftMax and FS-KNN. We also try three score fusion methods, including FS-KNN+S, FS-KNN*S, and SIRC, but these methods are general score fusion methods, while our method is specifically designed for UOSR, so our FS-KNNS surpass them. Our FS-KNNS is totally based on the existing model trained by the classical cross-entropy loss, so there is no extra effort and no outlier data during training, and our FS-KNNS still has better performance than all outlier exposure methods and achieves state-of-the-art UOSR performance, as shown in Fig. 1 (b) and 16. Note that the best choice for OSR (FS-KNN or SSD) may not be the best choice for UOSR (FS-KNNS), as their expectation of InC/InW is contradictory, which shows the necessity of our proposed few-shot UOSR.

# 6 CONCLUSION

Rejecting both InW and OoD samples during evaluation has very practical value in most real-world applications, but UOSR is a newly proposed problem and lacks comprehensive analysis and research, so we look deeper into the UOSR problem in this work. We first demonstrate the uncertainty distribution of almost all existing OSR methods is actually closer to the expectation of UOSR than OSR, and then analyze two training settings including pre-training and outlier exposure. Finally, we introduce a new evaluation setting into UOSR, which is few-shot UOSR, and we propose a FS-KNNS method that achieves state-of-the-art performance under all settings.

**Acknowledgement:** This work is supported by Alibaba Group through Alibaba Research Intern Program. Thanks to Xiang Wang and Zhiwu Qing who gave constructive comments on this work.

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

## A    RELATION BETWEEN MODEL CALIBRATION AND UOSR

Different from Selective Prediction (SP), Anomaly Detection (AD), OSR, and UOSR, Model Calibration (MC) is not an uncertainty ordinal ranking problem. In other words, all other settings expect the uncertainty of a test sample to be either 0 or 1, while MC is not. A perfect calibrated model should meet (Guo et al., 2017)

$$P(\hat{y} = y | f(x) = p) = p \quad \forall p \in [0, 1] \tag{3}$$

where $p$ is the confidence or the negative version of uncertainty $p = 1 - u$. For example, given 100 test samples whose confidence scores are all 0.8, then the model is perfectly calibrated if 80% samples are correctly classified. In this case, 20% samples with confidence 0.8 is consistent with the requirement of a perfect calibrated model, but the confidence is supposed to be either 0 or 1 in other settings. The performance of MC is evaluated by ECE, and the readers may refer to (Guo et al., 2017) for the formal definition. Smaller ECE means better calibrated model. We provide 5 examples to illustrate the relation between the performance of UOSR and MC in Fig. 6. Note that MC does not consider OoD samples during evaluation, so we also do not involve OoD samples in Fig. 6. Therefore, UOSR in Fig. 6 is also equal to SP. From (f) we can see that the performance of UOSR and model calibration is not perfectly positively correlated, *i.e.*, the best case for model calibration (c) is not the best case for UOSR and vice versa (e).

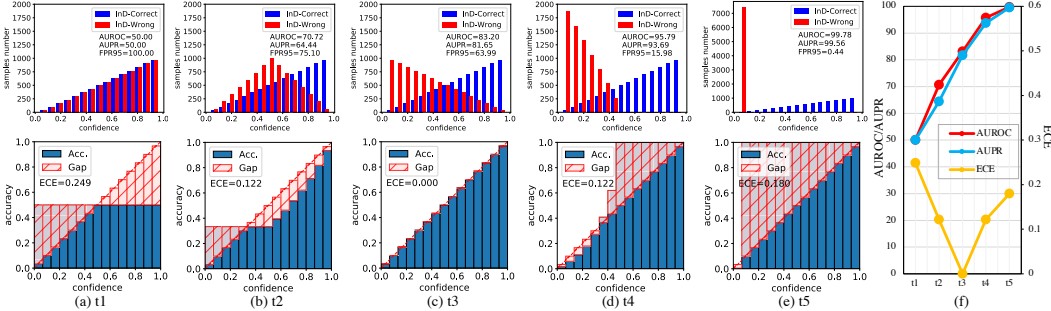

Figure 6: We provide 5 samples in (a)-(e), where we keep the confidence distribution of InC and change the confidence distribution of InW samples. The evaluation metrics of UOSR are AUROC and AUPR, and ECE is for the MC.

## B    TEMPERATURE SCALING FOR UOSR

Temperature scaling is a convenient and effective method for model calibration (Guo et al., 2017). We study how this method influences the UOSR performance. The experiments are conducted under R50 backbone while InD and OoD datasets are CIFAR100 and TinyImageNet, respectively. The quantitative results are in Table 8. The uncertainty distribution under different temperatures $T$ are in Fig. 7 and 8.

Table 8: UOSR and MC performance under different temperatures $T$.

| | w/o pre-training | | | | | | w/ pre-training | | | | |
|---|---|---|---|---|---|---|---|---|---|---|---|
| | ECE↓ | AURC↓ | | AUROC↑ | | | ECE↓ | AURC↓ | | AUROC↑ | |
| $T$ | MC | UOSR | UOSR | OSR | InC/InW | InC/OoD | MC | UOSR | UOSR | OSR | InC/InW | InC/OoD |
| 0.1 | 0.247 | 355.69 | 66.58 | 62.06 | 66.96 | 66.48 | 0.128 | 257.20 | 69.07 | 66.67 | 68.49 | 69.15 |
| 0.5 | 0.207 | 351.39 | 84.10 | 74.87 | 85.01 | 83.86 | 0.106 | 257.26 | 88.66 | 83.71 | 88.61 | 88.67 |
| 1 | 0.146 | 358.31 | 85.57 | 76.90 | **85.18** | 85.67 | 0.081 | 260.14 | 90.51 | 85.93 | **89.58** | 90.64 |
| 2 | **0.119** | 352.60 | 86.79 | 79.19 | 84.62 | 87.35 | **0.018** | 250.99 | 92.40 | 89.09 | 89.08 | 92.85 |
| 5 | 0.344 | 351.45 | 87.05 | 79.94 | 84.05 | 87.84 | 0.523 | **249.00** | **92.96** | 90.86 | 86.00 | 93.90 |
| 10 | 0.256 | 351.39 | 87.08 | 80.04 | 83.94 | 87.90 | 0.509 | 249.55 | 92.93 | 91.02 | 85.22 | 93.97 |
| 20 | 0.197 | **351.38** | **87.09** | **80.08** | 83.89 | **87.92** | 0.414 | 249.73 | 92.92 | **91.07** | 84.99 | **93.99** |

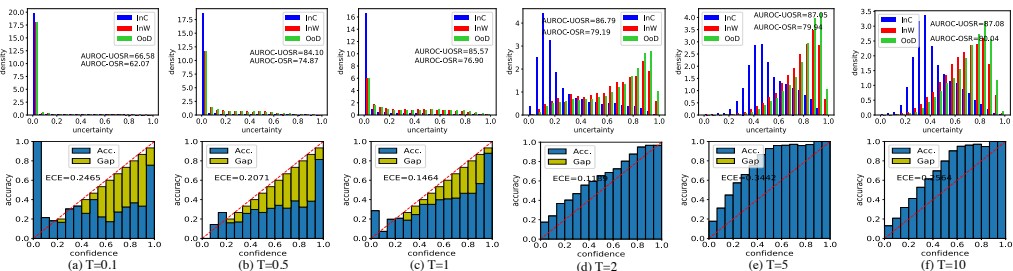

Figure 7: Uncertainty distribution under different temperatures $T$ without pre-training.

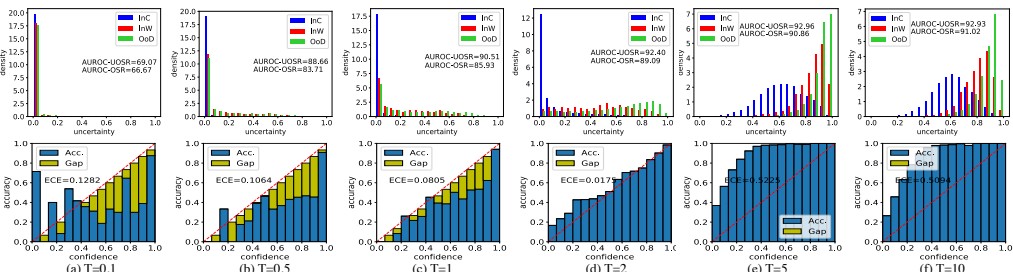

Figure 8: Uncertainty distribution under different temperature $T$ with pre-training.

From Table 8 we can see that the optimal $T$ for MC ($T = 2$) is not the best case for UOSR. When $T$ grows, the InC/OoD discrimination increases, but the InC/InW discrimination drops. Therefore, the OSR performance keeps improving with larger $T$, but the UOSR may not benefit from larger $T$ because of lower InC/InW discrimination. For example, the best $T$ for UOSR with pre-training is 5 rather than 20. But in general, temperature scaling is a simple and useful technique for both MC and UOSR, as $T = 2$ has better MC and UOSR performance than $T = 1$ (without temperature scaling). The only drawback of temperature scaling is it needs the validation set to determine the optimal $T$.

## C  EXISTING OSR METHODS ARE MORE SUITABLE FOR UOSR

In addition to Fig. 2, we provide the results of VGG13 backbone in the image domain and I3D backbone in the video domain. Fig. 9 further proves the finding that existing OSR methods are actually more suitable for UOSR holds for differnt model architectures and domains.

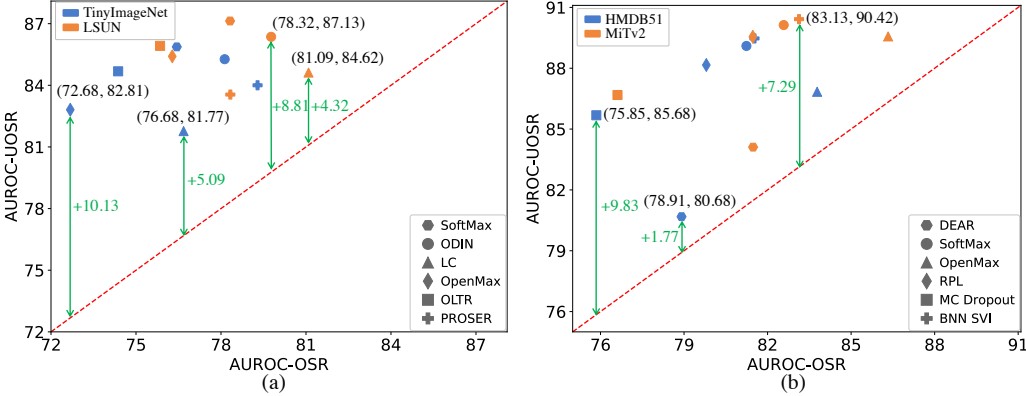

Figure 9: (a) and (b) are conducted using the VGG13 and I3D backbone in the image and video domain respectively. InD datasets are CIFAR100 and UCF101 for (a) and (b), and OoD datasets are shown with different colors.

## D   FEATURE AND UNCERTAINTY SCORE BEHAVIOR

To deeply understand why the InW samples share similar uncertainty distribution with OoD samples rather than InC samples, we analyze the feature representation and uncertainty score behavior in this section. Surprisingly, we find the InW samples have more similar features with InC samples than OoD samples, which is opposite from the uncertainty score behavior.

**Features of InC/InW/OoD samples are separable.** We visualize the feature representations in Fig 10. We find the feature distribution of InC, InW and OoD samples follow a hierarchy structure. The InW samples surround the InC samples, and OoD samples are further far away and located at the outer edge of InW samples, such as three distributions of class A and B in Fig 10 (b). Therefore, the features of InC, InW, and OoD samples are separable from the feature representation perspective. To further testify this idea, we calculate the similarity between InC/InW/OoD samples and training samples of each class, as shown in Fig. 11. We can see that the InC samples are the most similar samples with training samples, and InW samples have smaller similarity, and OoD samples have the smallest similarity. In conclusion, features of InC/InW/OoD samples are in a hierarchy structure and distinguishable.

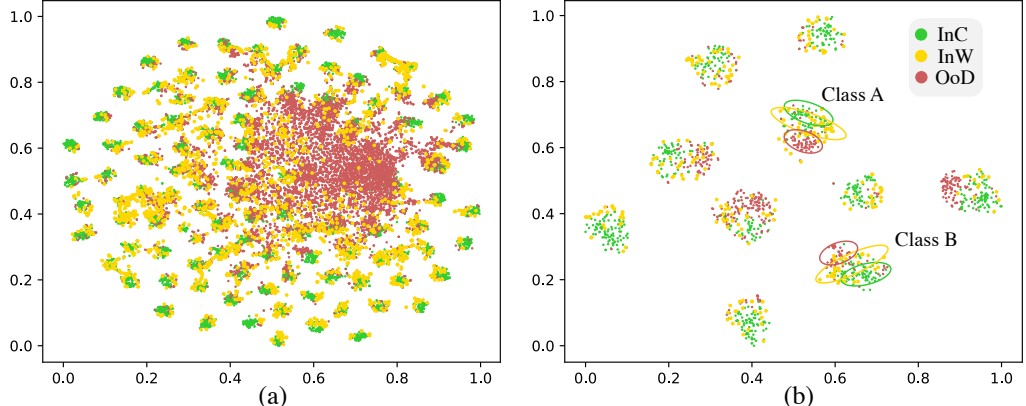

Figure 10: (a) and (b) are t-SNE visualization results of the whole test dataset and 10 classes.

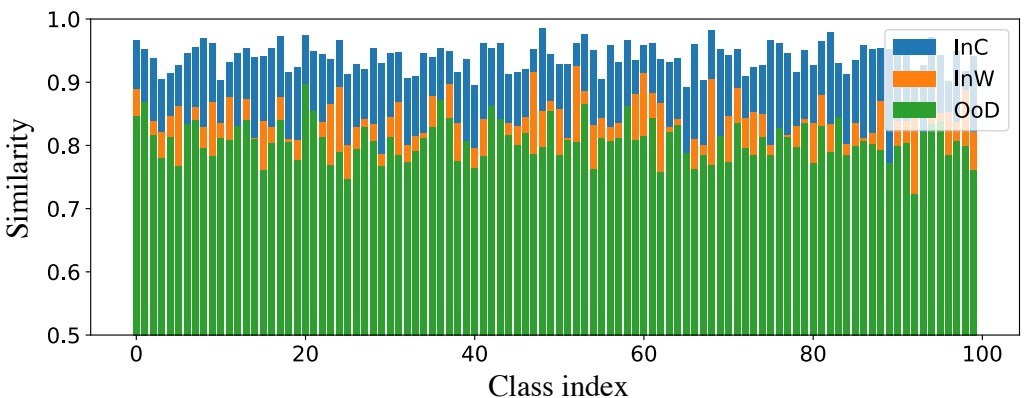

Figure 11: Similarity between with training samples of each class.

Table 9: We provide the feature similarity of InW/InC and InW/OoD, the mean of uncertainty score, and the AUROC of InW/InC and InW/OoD in this table.

|  | Similarity | | Uncertainty | | | Discrimination (AUROC) | |
|---|---|---|---|---|---|---|---|
|  | InW/InC | InW/OoD | InC | InW | OoD | InW/InC | InW/OoD |
| Feature space | 0.797 | 0.733 | 0.057 | 0.158 | 0.179 | 84.91 | 57.25 |
| Logit space | 0.718 | 0.571 | 0.314 | 0.537 | 0.583 | 83.36 | 57.95 |

**Features of InW samples are more similar with InC samples than OoD samples.** Our finding in Sec. 3 is that InW samples share similar uncertainty scores with OoD samples rather than InC samples, so we analyze whether the feature representation also follows the same behavior. We calculate the similarity between InW/InC samples and InW/OoD samples in the feature space and logit space. Then we provide the mean of uncertainty scores based on the KNN method and MaxLogit method, as well as the AUROC of InW/InC and InW/OoD to illustrate the uncertainty discrimination performance. In Table 9 we can see that the similarity of InW/InC is larger than InW/OoD (0.797-0.733), which means InW samples have more similar features with InC samples than OoD samples. However, the uncertainty scores of InW samples are more similar with OoD scores than InC samples (0.158/0.179-0.057), so that InW/OoD can not be distinguished very well like InW/InC (57.25-84.91). Therefore, we draw a very interesting conclusion that the feature behavior and uncertainty score behavior of InW/InC and InW/OoD are contradictory. InW samples are more similar to InC samples in the feature/logit space but more similar to OoD samples from the uncertainty score perspective.

Let us formulate this phenomenon mathematically. Suppose we have $x_c, x_w, x_o$ which represents the feature of an InC, InW, and OoD sample, respectively. From Table 9 we know that

$$sim(x_w, x_c) > sim(x_w, x_o), \tag{4}$$

where $sim$ refers to the similarity. Then, we have an uncertainty estimation function $f$ to measure the uncertainty $u$ of a sample based on the features, so $u_c = f(x_c), u_w = f(x_w), u_o = f(x_o)$. Based on our finding in Sec. 3 that InW samples share similar uncertainty distribution with OoD samples rather than InC samples, we have

$$sim(u_w, u_c) < sim(u_w, u_o), \text{or } sim(f(x_w), f(x_c)) < sim(f(x_w), f(x_o)). \tag{5}$$

Comparing Eq. 5 and Eq. 4 we find that the uncertainty estimation function $f$ changes the similarity relationship between InW/InC and InW/OoD.

Let us give a toy example of how $f$ changes the similarity relationship. Suppose the logit space is under 2 dimensions $(s, t)$, and $x_c = (2, 2), x_w = (1, 1), x_o = (1, 3)$. The similarity is measured with Euclidean distance, and in this case $x_w$ is closer to $x_c$, so Eq. 4 holds for this example. If $f(x) = -s$ like Fig. 12 (a), then $u_c = -2, u_w = -1, u_o = -1$. In this case, $u_w = u_o > u_c$. So the uncertainty score of InW sample is similar with OoD sample instead of InC sample. This example illustrates why an InW sample has a similar feature to an InC sample, but has a similar uncertainty score with OoD sample. This is how existing uncertainty estimation methods work, as the results in Sec. 3 and Sec. 4 show that InW samples have similar uncertainty distribution with OoD samples. This kind of method is suitable for UOSR problem where InW and OoD samples are supposed to be rejected at the same time.

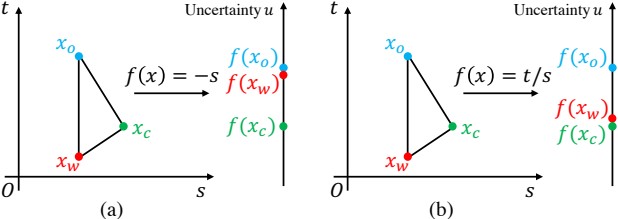

Figure 12: (a): $x_w$ is close to $x_c$ in $(s, t)$ space, but $f(x_w)$ is close to $f(x_o)$ in uncertainty space; (b) $x_w$ is close to $x_c$ in $(s, t)$ space, and $f(x_w)$ is also close to $f(x_c)$ in uncertainty space.

We provide another uncertainty estimation function $f$ in Fig. 12 (b), where $f(x) = t/s$. In this case, $u_w = u_c = 1 < u_o = 3$, so the uncertainty score of the InW sample is similar to the InC sample instead of the OoD sample. This is the ideal case for the traditional OSR problem to reject OoD samples and accept InC and InW samples.

**Conclusion.** In this section, we first find that the features of InC, InW and OoD samples are separable, and show that InW samples have more similar features to InC samples than OoD samples, which seems contradictory with the finding in Sec. 3 that InW samples share more similar uncertainty scores with OoD samples than InC samples. Then we give a toy example to illustrate that the uncertainty estimation method is the reason for this phenomenon. Although existing uncertainty estimation methods are designed to behave like Fig. 12 (b) for the OSR problem, they actually work as Fig. 12 (a) which is more suitable for the UOSR problem.

# E  IMPLEMENTATION DETAILS

## E.1  DEFINITION OF AURC

We follow (Kim et al., 2021) to give the definition of AURC here. Suppose we have a dataset $\mathcal{D} = \{(\mathbf{x}_i, y_i)\}_{i=1}^n$, where $\mathbf{x}_i$ and $y_i$ denote the training sample and corresponding closed-set label. Then, we use a confidence estimation method (methods in Table 5) to assign a confidence score $\kappa(\mathbf{p}_i)$ for each $\mathbf{x}_i$. Given a threshold $\theta$, the sample coverage is computed as

$$c(\theta) = \frac{|\mathcal{S}_\theta|}{|\mathcal{D}|}$$

where $|\cdot|$ is the number of samples in a set. Then the risk $r(\theta)$ is computed as

$$r(\theta) = \frac{\sum_{(\mathbf{x}_i, y_i) \in \mathcal{S}_\theta} \mathbb{1}(\hat{y}_i \neq y_i)}{|\mathcal{S}_\theta|}$$

where $\mathbb{1}(\cdot)$ and $\hat{y}_i$ denote the indicator function the predicted class respectively. So we have $(c(\theta), r(\theta))$ pairs which can consist of a risk-coverage curve (RC-curve). The AURC is the area under the RC-curve.

## E.2  IMAGE DOMAIN

**w/o pre-training.** When we train the model from scratch, we find that setting the base learning rate as 0.1 and step-wisely decayed by 10 every 24000 steps with totally 120000 steps can achieve good enough closed-set performance. We use a linear warmup strategy to warmup the training in the first 500 steps. We use SGD with momentum, batch size 128 for all models.

**w/ ImageNet-21k pre-training.** When we fine tune the model with ImageNet pretrained model from (Kolesnikov et al., 2020), we set the base learning rate as 0.003 and step-wisely decayed every 3000 steps with totally 10000 steps. We use a linear warmup strategy to warmup the training in the first 500 steps. We use SGD with momentum, batch size 512 for all models. For outlier exposure methods, the batch size of outlier data is set to 128.

**InD and OoD datasets.** We follow the datasets setting in (Liang et al., 2018). The training InD dataset is CIFAR-100, which contains 100 classes with 50000 training images and 10000 test images. The OoD datasets for open-set evaluation are TinyImageNet and LSUN. The TinyImageNet dataset contains 10000 test images from 200 different classes, we use the TinyImageNet (resize) in (Liang et al., 2018). The Large-scale Scene UNderstanding (LSUN) dataset consists of 10000 images of 10 different scenes categories like classroom, conference room, dining room, etc. We use LSUN (resize) from (Liang et al., 2018). The size of images in both InD and OoD datasets is 32 × 32. We use the same strategy in image preprocessing stage as (Kolesnikov et al., 2020), if the resolution of the training image is lower than 96 × 96, we use 128 × 128 image cropping technique and random horizontal mirroring followed by 160 × 160 image resize. Test images and OoD images are directly resized to 128 × 128.

**Outlier datasets.** We use 300K Random Images dataset from (Hendrycks et al., 2019b) as outlier dataset for those outlier exposure methods. The 300K Random Images dataset is a debiased dataset with real images scraped from online. According to (Hendrycks et al., 2019b), all images that belong to CIFAR classes and images with divisive metadata have been removed.

## E.3  VIDEO DOMAIN

**w/o pre-training.** When we train the model from scratch, we find that setting the base learning rate as 0.05 and step-wisely decayed every 160 epochs with totally 400 epochs can achieve good enough closed-set performance. The batch size is 256 for all methods.

**w/ Kinetics400 pre-training.** We follow (Bao et al., 2021) to set the base learning rate as 0.001 and step-wisely decayed every 20 epochs with totally 50 epochs. For those methods without outlier exposure, we fix the parameters of all Batch Normalization layers except the first one, and set the learning rate of the fully connected layer to be 10 times of the base learning rate. For those methods with outlier exposure, all parameters are updated with the same learning rate.

**InD and OoD datasets.** We follow the datasets setting in (Bao et al., 2021). The training InD dataset is UCF101, which contains 101 classes with 9537 training samples and 3783 test samples. The OoD datasets for open-set evaluation are HMDB51 and MiT-v2. We use the test sets of them which contain 1530 samples and 30500 samples respectively. For UCF101 and HMDB51, we follow the MMAction (Yue Zhao, 2019) to use the split 1 for training and evaluation, which is the same with (Bao et al., 2021). Note that in (Bao et al., 2021), they find some classes in HMDB51 overlap with those in UCF101 but they do not clean them. We remove the overlapping classes in UCF101 and HMDB51 so that OoD data does not contain any samples of InD classes. The classes we remove in HMDB51 and the corresponding same classes in UCF101 are in Table 10.

Table 10: Overlapping classes in HMDB51 and UCF101.

| HMDB51 | 35, Shoot bow | 29, Push up | 15, Golf | 26, Pull up |
|---|---|---|---|---|
| UCF101 | 2, Archery | 71, PushUps | 32, GolfSwing | 69, PullUps |
| HMDB51 | 30, Ride bike | 34, Shoot ball | 43, Swing baseball | 31, Ride horse |
| UCF101 | 10, Biking | 7, Basketball | 6, BaseballPitch | 41, HorseRiding |

**Outlier datasets.** We use Kinetics400 as our outlier datasets for those outlier exposure methods. To ensure that the classes of outlier data is not overlapping with InD data and OoD data, we remove corresponding classes in Kinetics400. The overlapping classes between Kinetics400 and UCF101/HMDB51 are too many to be listed here (129 overlapping classes). The available training sample ID list of Kinetics400 and all codes will be public. We pick up 271 classes from Kinetics400 and 25 samples in each class as outlier data.

## F    DETAILED UNCERTAINTY ANALYSIS OF TABLE 5

Table 11 is the complementary analysis of Table 5, which further illustrates that outlier data can improve InC/OoD discrimination but may not be helpful for InC/InW discrimination.

Table 11: Uncertainty distribution analysis in image domain with ResNet50. Pre-training is not used. OoD dataset: TinyImageNet. AUORC (%) is reported.

| Methods | OD | InC/OoD | InC/InW | InW/OoD | OSR | UOSR |
|---|---|---|---|---|---|---|
| ODIN | N | 88.35 | 80.76 | 64.36 | 81.65 | 86.69 |
| LC | N | 84.60 | 82.58 | 55.14 | 76.37 | 84.16 |
| OpenMax | N | 85.16 | 85.96 | 51.27 | 76.23 | 85.33 |
| OLTR | N | 85.99 | 85.74 | 52.22 | 77.10 | 85.94 |
| SoftMax | N | 84.69 | 85.68 | 50.64 | 75.59 | 84.90 |
| SoftMax(OE) | R | 90.04 | 84.00 | 68.54 | 84.35 | 88.78 |
| ARPL | N | 88.76 | 85.77 | 58.09 | 80.49 | 88.13 |
| ARPL+CS | R | 89.36 | 81.40 | 65.32 | 82.81 | 87.65 |
| MCD(Dropout) | N | 84.15 | 74.17 | 63.15 | 79.21 | 82.25 |
| MCD | R | 87.47 | 85.23 | 61.39 | 79.88 | 86.97 |
| PROSER | G | 87.04 | 77.84 | 62.57 | 79.23 | 84.82 |
| PROSER (EB) | R | 88.36 | 77.88 | 65.60 | 81.95 | 86.06 |
| VOS | G | 87.51 | 83.38 | 58.17 | 79.72 | 86.65 |
| VOS | R | 91.44 | 84.41 | 70.32 | 85.78 | 89.96 |

## G    UOSR BENCHMARK UNDER TRADITIONAL OSR SETTING

We evaluate the UOSR and OSR performance under the traditional OSR dataset setting (Vaze et al., 2022), where a part of data within one dataset is regarded as InD and the remaining data is regarded as OoD. The experiments are conducted under the most challenging TinyImageNet dataset and the results are in Table 12. We can see that the UOSR performance is still higher than OSR performance for most methods, which means InW samples share more similar uncertainty distribution with InC samples than OoD samples. Surprisingly, we find the InW/OoD AUORC of ODIN method achieves 84.32, which means InW and OoD samples can be well distinguishable. This proves our claim in Appendix D that the features of InC/InW/OoD samples are separable and it is possible to find a proper uncertainty estimation method to distinguish these three groups of data.

Table 12: Unified open-set recognition benchmark in the image domain under the traditional OSR dataset setting. All methods are conducted under the R50 model. Dataset is TinyImageNet. $\dagger, \ddagger, \diamond$ refer to OSR-based, SP-based, UOSR-based methods. Pre-training weights are used.

| Methods | UOSR | | | OSR | InC/InW | InC/OoD | InW/OoD |
|---|---|---|---|---|---|---|---|
| | Acc.↑ | AURC↓ | AUROC↑ | AUROC↑ | AUROC↑ | AUROC↑ | AUROC↑ |
| SoftMax$^\dagger$ | 87.40 | 730.10 | 94.06 | 90.59 | 91.02 | 94.11 | 66.21 |
| ODIN$^\dagger$ | 87.40 | 724.90 | 96.62 | 95.21 | 85.50 | 96.78 | 84.32 |
| LC$^\dagger$ | 87.40 | 756.49 | 88.02 | 84.32 | 81.27 | 88.11 | 58.05 |
| OpenMax$^\dagger$ | 86.60 | 731.73 | 94.49 | 90.25 | 91.77 | 94.53 | 62.59 |
| OLTR$^\dagger$ | 87.60 | 731.93 | 93.69 | 90.09 | 89.96 | 93.74 | 64.34 |
| PROSER$^\dagger$ | 86.90 | 750.26 | 92.29 | 90.21 | 77.01 | 92.51 | 74.91 |
| BCE$^\ddagger$ | 87.60 | 732.79 | 93.74 | 90.46 | 89.63 | 93.80 | 66.92 |
| TCP$^\ddagger$ | 87.70 | 731.66 | 94.20 | 90.54 | 90.20 | 94.25 | 64.06 |
| DOCTOR$^\ddagger$ | 87.40 | 731.69 | 93.97 | 90.54 | 90.25 | 94.02 | 66.40 |
| SIRC(MSP, $\|z\|_1$)$^\diamond$ | 87.40 | 731.48 | 94.02 | 90.53 | 91.04 | 94.06 | 66.06 |
| SIRC(MSP, Res.)$^\diamond$ | 87.40 | 729.96 | 94.77 | 91.71 | 91.19 | 94.82 | 70.13 |
| SIRC($-\mathcal{H}$, $\|z\|_1$)$^\diamond$ | 87.40 | 730.12 | 94.74 | 91.65 | 91.24 | 94.79 | 69.89 |
| SIRC($-\mathcal{H}$, Res.)$^\diamond$ | 87.40 | 731.34 | 94.06 | 90.59 | 91.02 | 94.10 | 66.22 |

# H  UOSR BENCHMARK UNDER DIFFERENT TRAINING SET

We provide the UOSR evaluation results in Table 13 when InD and OoD datasets are TinyImageNet and CIFAR100 respectively. Pre-training weights are used. TinyImageNet has 200 classes in the training set which is more diverse than CIFAR100. We can see that training the model on Tiny-ImageNet is more challenging than CIFAR100, as the Acc. of CIFAR100 is 86.44 and Acc. of TinyImageNet only reaches 77.02. In this way, the impact of InW samples becomes further huge. For example, the performance gap between UOSR and OSR for the SoftMax method is 8.95 when InD dataset is TinyImageNet, and this value is only 0.34 when InD dataset is CIFAR100. So the InW samples are more important for the performance when InD dataset is difficult, as lower closed-set Acc. means more InW samples.

Table 13: Unified open-set recognition benchmark in the image domain. All methods are conducted under the R50 model. InD and OoD Dataset are TinyImageNet and CIFAR100 respectively. $\dagger, \ddagger, \diamond$ refer to OSR-based, SP-based, UOSR-based methods. Pre-training weights are used.

| Methods | UOSR | | | OSR | InC/InW | InC/OoD | InW/OoD |
|---|---|---|---|---|---|---|---|
| | Acc.↑ | AURC↓ | AUROC↑ | AUROC↑ | AUROC↑ | AUROC↑ | AUROC↑ |
| SoftMax$^\dagger$ | 77.02 | 340.70 | 86.22 | 77.27 | 88.83 | 85.62 | 49.27 |
| ODIN$^\dagger$ | 77.23 | 359.57 | 84.01 | 75.32 | 86.53 | 83.43 | 47.81 |
| LC$^\dagger$ | 77.23 | 385.15 | 79.76 | 71.85 | 83.21 | 78.98 | 47.69 |
| OpenMax$^\dagger$ | 76.90 | 340.21 | 86.53 | 77.09 | 89.63 | 85.81 | 48.04 |
| OLTR$^\dagger$ | 77.00 | 341.60 | 86.04 | 76.75 | 89.02 | 85.36 | 47.92 |
| PROSER$^\dagger$ | 75.93 | 392.91 | 80.50 | 74.65 | 79.20 | 80.50 | 55.20 |
| BCE$^\ddagger$ | 76.91 | 339.43 | 86.40 | 76.83 | 89.64 | 85.66 | 47.44 |
| TCP$^\ddagger$ | 77.82 | 336.61 | 86.48 | 77.47 | 89.72 | 85.76 | 48.38 |
| DOCTOR$^\ddagger$ | 77.23 | 339.02 | 86.69 | 77.62 | 89.86 | 85.96 | 49.30 |
| SIRC(MSP, $\|z\|_1$)$^\diamond$ | 77.03 | 337.28 | 86.82 | 78.86 | 88.78 | 86.37 | 53.65 |
| SIRC(MSP, Res.)$^\diamond$ | 77.03 | 316.14 | 90.66 | 87.00 | 88.82 | 91.08 | 73.31 |
| SIRC($-\mathcal{H}$, $\|z\|_1$)$^\diamond$ | 77.03 | 333.73 | 87.60 | 80.13 | 89.00 | 87.28 | 56.19 |
| SIRC($-\mathcal{H}$, Res.)$^\diamond$ | 77.03 | 311.98 | 91.39 | 88.20 | 89.02 | 91.94 | 75.67 |

# I UOSR BENCHMARK UNDER SSB DATASETS

(Vaze et al., 2022) proposed the Semantic Shift Benchmark (SSB) which compose several fine-grained datasets, including CUB, Standford Cars, FGCV-Aircraft, and a part of ImageNet. OSR in SSB is more challenging as OoD samples share the same coarse labels with InD samples, and only have some minor differences in the fine-grained properties. We evaluate the UOSR performance on CUB and FGCV-Aircraft and provide the results of EASY and HARD modes. Pre-training weights are used for better performance. From Table 14 and 15 we can see the UOSR and OSR performance are higher under the EASY mode compared to HARD mode as expected, since the OoD samples are more similar with InD samples in the HARD mode. Our conclusion that InW samples share similar uncertainty with OoD samples still holds, as the AUROC of InW/OoD is close to 50 and much lower than InC/OoD and InC/InW.

Table 14: Unified open-set recognition benchmark in the image domain. All methods are conducted under the R50 model. Dataset is CUB-200-2011. $\dagger, \ddagger, \diamond$ refer to OSR-based, SP-based, UOSR-based methods. Pre-training weights are used. EASY/HARD

| Methods | UOSR | | | OSR | InC/InW | InC/OoD | InW/OoD |
| | Acc.↑ | AURC↓ | AUROC↑ | AUROC↑ | AUROC↑ | AUROC↑ | AUROC↑ |
|---|---|---|---|---|---|---|---|
| SoftMax$^\dagger$ | 91.78 | 77.69/120.46 | 92.79/84.31 | 90.33/78.78 | 90.56 | 93.37/82.81 | 56.34/33.73 |
| ODIN$^\dagger$ | 91.61 | 86.89/157.09 | 91.20/77.37 | 91.45/73.11 | 82.42 | 93.52/76.14 | 68.90/40.09 |
| LC$^\dagger$ | 91.61 | 78.34/121.66 | 92.66/84.35 | 89.86/78.63 | 91.15 | 93.06/82.69 | 54.95/34.38 |
| OpenMax$^\dagger$ | 91.30 | 78.07/119.61 | 92.87/85.43 | 90.01/79.78 | 91.14 | 93.35/83.98 | 54.99/35.67 |
| OLTR$^\dagger$ | 91.33 | 80.38/118.50 | 92.43/85.30 | 89.42/79.72 | 90.66 | 92.91/83.95 | 52.65/35.19 |
| PROSER$^\dagger$ | 91.33 | 79.39/128.14 | 92.50/83.81 | 90.32/78.53 | 89.31 | 93.37/82.42 | 58.21/37.60 |
| BCE$^\ddagger$ | 91.50 | 79.75/122.19 | 92.24/84.76 | 89.34/79.27 | 90.64 | 92.67/83.31 | 53.43/35.80 |
| TCP$^\ddagger$ | 92.06 | 77.31/116.93 | 92.55/84.85 | 90.28/79.92 | 90.07 | 93.17/83.64 | 56.71/36.78 |
| DOCTOR$^\ddagger$ | 91.61 | 78.09/121.61 | 92.76/84.37 | 90.09/78.68 | 91.13 | 93.18/82.71 | 56.26/34.70 |
| SIRC(MSP, $\|z\|_1$)$^\diamond$ | 91.78 | 78.04/119.42 | 92.72/84.46 | 90.19/78.95 | 90.59 | 93.27/83.00 | 55.78/33.70 |
| SIRC(MSP, Res.)$^\diamond$ | 91.78 | 77.69/120.46 | 92.79/84.31 | 90.33/78.78 | 90.56 | 93.37/82.81 | 56.33/33.74 |
| SIRC($-\mathcal{H}$, $\|z\|_1$)$^\diamond$ | 91.78 | 76.97/119.40 | 93.11/84.52 | 91.06/83.08 | 90.52 | 93.78/83.08 | 60.65/34.78 |
| SIRC($-\mathcal{H}$, Res.)$^\diamond$ | 91.78 | 76.65/120.65 | 93.18/84.33 | 91.19/78.91 | 90.47 | 93.88/82.86 | 61.20/34.81 |

Table 15: Unified open-set recognition benchmark in the image domain. All methods are conducted under the R50 model. Dataset is Fine-Grained Visual Classification of Aircraft (FGVC-Aircraft). $\dagger, \ddagger, \diamond$ refer to OSR-based, SP-based, UOSR-based methods. Pre-training weights are used. EASY/HARD

| Methods | UOSR | | | OSR | InC/InW | InC/OoD | InW/OoD |
| | Acc.↑ | AURC↓ | AUROC↑ | AUROC↑ | AUROC↑ | AUROC↑ | AUROC↑ |
|---|---|---|---|---|---|---|---|
| SoftMax$^\dagger$ | 85.61 | 129.71/127.18 | 89.46/82.46 | 84.24/72.85 | 88.54 | 89.80/79.08 | 51.18/35.75 |
| ODIN$^\dagger$ | 85.25 | 152.17/173.15 | 87.11/75.75 | 84.92/68.36 | 80.04 | 89.72/73.32 | 57.17/39.70 |
| LC$^\dagger$ | 85.25 | 134.44/144.29 | 88.98/80.15 | 83.55/69.74 | 87.58 | 89.50/75.93 | 49.13/33.95 |
| OpenMax$^\dagger$ | 86.69 | 123.31/123.86 | 90.13/82.10 | 85.63/73.20 | 88.12 | 90.80/79.01 | 51.99/35.36 |
| OLTR$^\dagger$ | 85.97 | 128.01/124.92 | 89.86/82.79 | 85.11/73.45 | 88.57 | 90.31/79.66 | 53.26/35.35 |
| PROSER$^\dagger$ | 85.97 | 124.15/137.70 | 90.73/80.40 | 86.92/70.44 | 87.31 | 91.93/76.67 | 56.24/32.22 |
| BCE$^\ddagger$ | 85.37 | 134.56/127.05 | 88.75/82.37 | 84.05/73.93 | 86.49 | 89.58/80.06 | 51.75/38.17 |
| TCP$^\ddagger$ | 85.25 | 132.09/131.80 | 89.30/82.19 | 83.60/72.12 | 88.41 | 89.63/78.66 | 48.73/34.30 |
| DOCTOR$^\ddagger$ | 85.25 | 133.93/144.07 | 89.16/80.24 | 83.79/69.80 | 87.68 | 89.71/76.00 | 49.58/33.96 |
| SIRC(MSP, $\|z\|_1$)$^\diamond$ | 85.61 | 130.50/126.36 | 89.24/82.57 | 83.86/73.04 | 88.52 | 89.49/79.27 | 50.31/35.98 |
| SIRC(MSP, Res.)$^\diamond$ | 85.61 | 129.71/127.18 | 89.46/82.46 | 84.24/72.85 | 88.54 | 89.80/79.08 | 51.17/35.76 |
| SIRC($-\mathcal{H}$, $\|z\|_1$)$^\diamond$ | 85.61 | 128.97/125.59 | 89.71/82.85 | 84.70/73.60 | 88.55 | 90.13/79.70 | 52.40/37.35 |
| SIRC($-\mathcal{H}$, Res.)$^\diamond$ | 85.61 | 128.04/126.58 | 89.97/82.69 | 85.16/73.35 | 88.52 | 90.49/79.46 | 53.46/37.01 |

## J UOSR BENCHMARK AND FEW-SHOT RESULTS OF VGG13

We provide the UOSR benchmark and few-shot results of VGG13 in Table 16 and 17 respectively. Our FS-KNNS still achieves the best performance under all settings.

Table 16: Unified open-set recognition benchmark in the image domain. All methods are conducted under the VGG13 model. InD dataset is CIFAR100 while OoD dataset is TinyImageNet. †, ‡, ◊ refer to OSR-based, SP-based, UOSR-based methods. OD: use Outlier Data in training.

| | | w/o pre-training | | | | w/ pre-training | | | |
| | | UOSR | | | OSR | UOSR | | | OSR |
| Methods | OD | Acc.↑ | AURC↓ | AUROC↑ | AUROC↑ | Acc.↑ | AURC↓ | AUROC↑ | AUROC↑ |
|---|---|---|---|---|---|---|---|---|---|
| SoftMax† | ✗ | 75.07 | 341.06 | 85.87 | 76.44 | 74.69 | 311.70 | 91.01 | 85.13 |
| ODIN† | ✗ | 75.07 | 346.70 | 85.27 | 78.13 | 74.69 | 314.80 | 91.24 | 88.90 |
| LC† | ✗ | 75.07 | 365.96 | 81.77 | 76.68 | 74.69 | 339.96 | 88.27 | 88.95 |
| OpenMax† | ✗ | 74.52 | 368.58 | 82.81 | 72.68 | 75.05 | 312.70 | 90.47 | 84.25 |
| OLTR† | ✗ | 73.80 | 365.61 | 84.68 | 74.37 | 74.52 | 306.73 | 92.12 | 87.08 |
| PROSER† | ✗ | 70.95 | 376.63 | 84.00 | 79.29 | 71.11 | 367.16 | 86.45 | 85.97 |
| BCE‡ | ✗ | 74.74 | 339.25 | 86.54 | 76.99 | 74.45 | 325.29 | 88.85 | 81.14 |
| TCP‡ | ✗ | 75.09 | 340.41 | 85.94 | 76.18 | 74.83 | 313.80 | 90.57 | 84.35 |
| DOCTOR‡ | ✗ | 75.07 | 340.69 | 85.95 | 76.72 | 74.69 | 310.29 | 91.37 | 85.97 |
| SIRC(MSP, $\|z\|_1$)◊ | ✗ | 75.07 | 343.44 | 85.33 | 75.63 | 74.69 | 313.22 | 90.67 | 84.59 |
| SIRC(MSP, Res.)◊ | ✗ | 75.07 | 336.87 | 86.77 | 78.23 | 74.69 | 309.67 | 91.46 | 86.14 |
| SIRC($-\mathcal{H}$, $\|z\|_1$)◊ | ✗ | 75.07 | 343.53 | 85.25 | 76.37 | 74.69 | 309.19 | 91.56 | 86.82 |
| SIRC($-\mathcal{H}$, Res.)◊ | ✗ | 75.07 | 335.71 | 86.93 | 79.13 | 74.69 | 305.34 | 92.39 | 88.34 |
| OE | ✓ | 71.71 | 312.44 | 93.71 | 89.76 | 73.38 | 306.72 | 93.34 | 92.77 |
| EB | ✓ | 74.19 | 340.66 | 87.67 | 85.73 | 72.76 | 340.81 | 89.40 | 89.79 |
| VOS | ✓ | 71.73 | 312.75 | 93.68 | 90.76 | 73.08 | 306.37 | 93.65 | 92.97 |
| MCD | ✓ | 70.45 | 316.57 | 94.21 | 91.54 | 72.08 | 300.87 | 95.40 | 94.26 |

Table 17: Results of few-shot UOSR in the image domain. Model is VGG13 with pre-training. InD and OoD datasets are CIFAR100 and TinyImageNet respectively.

| | 5-shot | | | | | 1-shot | | | | |
| | AURC↓ | AUROC↑ | | | | AURC↓ | AUROC↑ | | | |
| Methods | UOSR | UOSR | OSR | InC/OoD | InC/InW | UOSR | UOSR | OSR | InC/OoD | InC/InW |
|---|---|---|---|---|---|---|---|---|---|---|
| SoftMax | 311.68 | 91.00 | 85.13 | 92.10 | 86.68 | 311.68 | 91.00 | 85.13 | 92.10 | 86.68 |
| KNN | 306.04 | 93.04 | 91.61 | 95.75 | 82.32 | 306.04 | 93.04 | 91.61 | 95.75 | 82.32 |
| FS-KNN | 295.80 | 95.44 | 98.87 | **99.39** | 79.83 | 296.75 | 95.27 | 98.48 | 99.16 | 79.87 |
| SSD | 321.31 | 92.57 | **99.26** | 99.36 | 65.72 | 315.46 | 93.11 | **99.10** | **99.27** | 68.77 |
| FS-KNN+S | 296.42 | 94.22 | 92.15 | 96.50 | 85.20 | 296.67 | 94.20 | 92.15 | 96.48 | 85.16 |
| FS-KNN*S | 303.01 | 92.85 | 89.42 | 94.38 | **86.83** | 302.84 | 92.88 | 89.45 | 94.41 | **86.83** |
| SIRC | 287.91 | 95.71 | 96.33 | 98.00 | 86.66 | 288.12 | 95.68 | 96.30 | 98.00 | 86.53 |
| FS-KNNS | **285.85** | **95.95** | 96.52 | 98.30 | 86.66 | **287.02** | **95.75** | 95.96 | 98.04 | 86.67 |

## K UOSR UNDER NOISY OUTLIER EXPOSURE

When we introduce outlier data into the training process, the labels of some outlier data may be corrupted and become the labels of InD class. We call this kind of outlier data as noisy outlier data. We study how noisy outlier data influence the UOSR and OSR performance in this section. In addition, we use NGC (Wu et al., 2021) to find the noisy outlier data and correct them. The results are shown in Table 18. We can see that the closed-set Acc. gradually decreases with the growth of noise level when NGC is not used. This is natural as some noisy outlier data corrupt the InD data distribution. In contrast, the closed-set Acc. does not drop with the growth of noise level when NGC

is used. So NGC is very effective in finding corrupted samples. When the noise level is 0%, the UOSR performance with NGC is better than the performance without NGC (lower AURC value), meaning that other parts in NGC that are not related to label corruption, such as contrastive learning, are helpful for UOSR. Surprisingly, the performance of UOSR and OSR are relatively stable under different noise levels no matter we use NGC or not, compared to the clear performance drop of closed-set Acc. when NGC is not used. So the model is robust in open-set related performance when noisy outlier data is introduced. Our finding that InD samples share similar uncertainty scores with OoD samples still holds as AUROC of InW/OoD is close to 50.

Table 18: UOSR and OSR performance under noisy outlier data. InD dataset is CIFAR100 and outlier dataset is 300K Random Images. OoD dataset is TinyImageNet. Experiments are conducted with ResNet18 backbone.

| Noise level | NGC | UOSR Acc.↑ | UOSR AURC↓ | UOSR AUROC↑ | OSR AUROC↑ | InC/InW AUROC↑ | InC/OoD AUROC↑ | InW/OoD AUROC↑ |
|---|---|---|---|---|---|---|---|---|
| 0% | ✗ | 76.19 | 334.13 | 85.50 | 77.29 | 85.05 | 85.60 | 50.70 |
| 20% | ✗ | 72.15 | 343.46 | 87.84 | 77.53 | 87.34 | 87.98 | 50.45 |
| 40% | ✗ | 70.06 | 343.60 | 89.91 | 79.86 | 88.36 | 90.37 | 55.25 |
| 60% | ✗ | 69.46 | 346.62 | 90.07 | 79.79 | 88.76 | 90.47 | 55.52 |
| 80% | ✗ | 69.20 | 348.18 | 89.95 | 79.05 | 88.84 | 90.29 | 53.77 |
| 100% | ✗ | 68.05 | 356.35 | 89.97 | 78.31 | 89.63 | 90.08 | 53.23 |
| 0% | ✓ | 77.52 | 315.32 | 87.75 | 81.12 | 85.61 | 88.23 | 56.61 |
| 20% | ✓ | 77.50 | 316.86 | 87.81 | 80.46 | 86.88 | 88.02 | 54.40 |
| 40% | ✓ | 76.87 | 321.66 | 87.58 | 79.48 | 87.15 | 87.68 | 52.23 |
| 60% | ✓ | 77.07 | 319.58 | 87.91 | 79.88 | 87.43 | 88.02 | 52.53 |
| 80% | ✓ | 77.12 | 317.90 | 88.17 | 80.14 | 87.41 | 88.34 | 52.49 |
| 100% | ✓ | 77.38 | 328.85 | 86.24 | 78.00 | 86.46 | 86.20 | 49.94 |

## L ABLATION STUDY OF FEW-SHOT UOSR

**Analyze of $K$.** Similar with (Sun et al., 2022), we find the value of $K$ influences the performance a lot. The ablation study of $K$ is shown in Fig. 13. It shows that the best $K$ for InC/OoD discrimination is between 3 and 5, and drops quickly after 7. In contrast, the InC/InW performance keeps increasing until 20 and then drops. The overall UOSR performance achieves the best when $K = 5$.

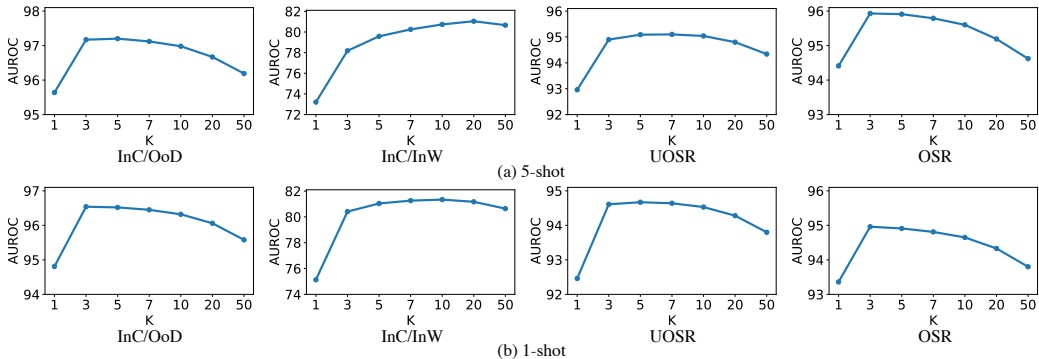

Figure 13: Ablation study of $K$ used in FS-KNN. The backbone is ResNet50.

**Analyze of $\alpha$ and $\lambda_{knns}$.** Two hyper parameters of FS-KNNS including $\alpha$ and $\lambda_{knns}$ are significant for the performance. $\lambda_{knns}$ is a threshold to determine when the weight of $u_1$ becomes large, *i.e.*, $\hat{u}_{fs-knns}$ is important when $u_1 > \lambda_{knns}$. $\alpha$ is to control the change rate of the weight. The ideal $\lambda_{knns}$ should be located between the uncertainty of InW and OoD samples as shown in Fig. 5 (a) and (c), so that the uncertainty of InW samples is still determined by SoftMax and the uncertainty of OoD samples is enlarged because of FS-KNN. However, which sample is InW or OoD is unknown during test, so we cannot determine the $\lambda_{knns}$ based on the uncertainty distribution of test samples. (Xia & Bouganis, 2022) proposed to determine the hyper parameters of SIRC based on the training

set, but we find that the uncertainty distribution $\hat{u}_{fs-knns}$ of training samples significantly different from the test InC samples, as shown in Fig. 14. Therefore, we seek help from the OoD reference

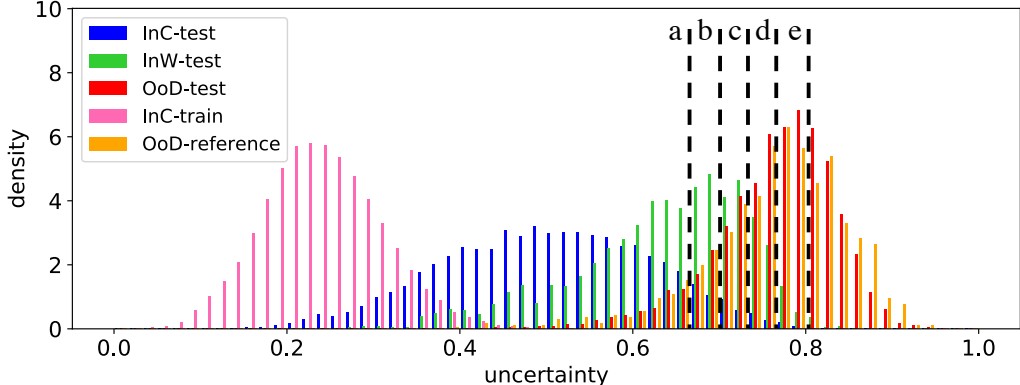

Figure 14: Uncertainty distribution of $\hat{u}_{fs-knns}$. InC-train samples have distinct uncertainty distribution with InC-test samples, but OoD reference samples share similar uncertainty distribution with OoD test samples. a to e correspond to the $\lambda_{knns}$ when $\beta = 1.5, 1, 0.5, 0, -0.5$ in Eq. 6.

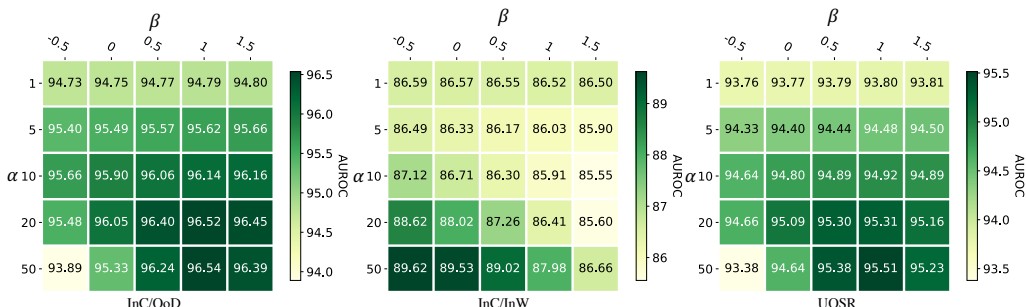

Figure 15: Ablation study of $\beta$ and $\alpha$ in Eq. 2 and 6.

samples, as we find their uncertainty distribution is extremely similar with OoD test samples. We calculate the mean $\overline{u}$ and standard deviation $\sigma$ of the $\hat{u}_{fs-knns}$ of OoD reference samples, and we aim to find a proper $\lambda_{knns}$ through:

$$\lambda_{knns} = \overline{u} - \beta \cdot \sigma \tag{6}$$

We draw several $\lambda_{knns}$ with different $\beta$ in Fig. 14 (a to e). A smaller $\lambda_{knns}$ or a larger $\beta$ means more samples will be influenced by the FS-KNN uncertainty, so that the $\hat{u}_{fs-knns}$ of more OoD test samples will be strengthened by the FS-KNN uncertainty which brings better InC/OoD performance, but meanwhile more InW samples will also be influenced by the FS-KNN uncertainty which brings worse InC/InW performance, as shown in Fig. 15. In other words, $\lambda_{knns}$ controls the trade-off between InC/OoD and InC/InW performance. Overall, $\beta = 1$ achieves the best UOSR performance which is a balanced result of InC/OoD and InC/InW trade-off. Larger $\alpha$ means the weight of FS-KNN uncertainty grows quickly when FS-KNN uncertainty is closed to $\lambda_{knns}$. We find that a smaller $\alpha$ makes the performance insensitive of $\lambda_{knns}$, and vice versa. So finally we pick $\alpha$ and $\beta$ as 50 and 1 as the optimal hyper parameters.

## M   FEW-SHOT UOSR IN THE VIDEO DOMAIN

Table 19: Results of few-shot UOSR in the video domain. Model is TSM with pre-training. InD and OoD datasets are UCF101 and HMDB51. AUORC (%) and AURC ($\times 10^3$) are reported.

| | 5-shot | | | | | 1-shot | | | | |
|---|---|---|---|---|---|---|---|---|---|---|
| | **AURC↓** | **AUROC↑** | | | | **AURC↓** | **AUROC↑** | | | |
| **Methods** | **UOSR** | **UOSR** | **OSR** | **InC/OoD** | **InC/InW** | **UOSR** | **UOSR** | **OSR** | **InC/OoD** | **InC/InW** |
| SoftMax | 66.55 | 93.66 | 91.44 | 93.46 | **94.99** | 66.55 | 93.66 | 91.44 | 93.46 | **94.99** |
| KNN | 73.73 | 93.38 | 92.99 | 94.11 | 88.44 | 73.73 | 93.38 | 92.99 | 94.11 | 88.44 |
| FS-KNN | 68.44 | 95.04 | **96.14** | **96.71** | 83.74 | 73.19 | 93.60 | **94.32** | 95.09 | 83.53 |
| SSD | 70.36 | 93.91 | 95.41 | 95.97 | 79.97 | 75.61 | 92.48 | 93.34 | 94.07 | 81.75 |
| FS-KNN+S | 65.64 | 95.44 | 94.69 | 96.13 | 90.78 | 68.98 | 94.54 | 93.66 | 95.14 | 90.48 |
| FS-KNN*S | 65.29 | 94.09 | 92.13 | 93.96 | **94.99** | 65.42 | 94.04 | 92.08 | 93.91 | 94.97 |
| SIRC | 62.41 | 95.11 | 93.79 | 95.13 | 94.94 | 63.11 | 94.88 | 93.49 | 94.88 | 94.84 |
| FS-KNNS | **60.00** | **96.19** | 95.37 | 96.40 | 94.82 | **62.65** | **95.35** | 94.26 | **95.48** | 94.49 |

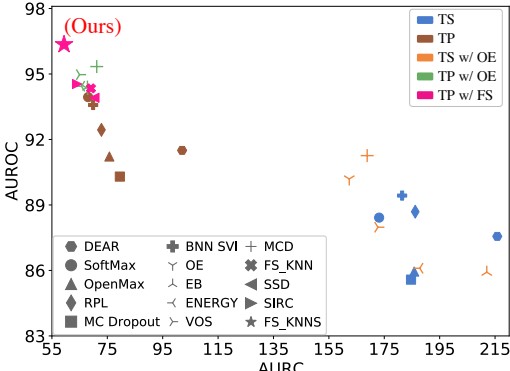

Figure 16: UOSR performance under all settings of TSM backbone in the video domain. OoD dataset is HMDB51.

