# OpenReview forum: "The Devil is in the Wrongly-classified Samples: Towards Unified Open-set Recognition"
_ICLR.cc/2023/Conference — ICLR 2023 poster_

### Official Review · Reviewer_sZtz · 2022-10-22

**Confidence:** 4
**Correctness:** 4
**Technical Novelty And Significance:** 2
**Empirical Novelty And Significance:** 4
**Recommendation:** 8

**Clarity, Quality, Novelty And Reproducibility:**

I believe the quality of this paper is high, with all experimental details clearly specified and claims substantiated.

However, I believe that the clarity of the paper could be substantially improved in the following ways.

1) The take home from the paper, that 'false positive' predictions tend to be misclassified closed-set samples, is not presented in a straightforward manner but typically in a more roundabout way (e.g 'the uncertainty distribution of almost all existing methods designed for OSR is actually closer to the expectation of UOSR than OSR'). This hurts readability.
2) As mentioned above, AURC should be clearly explained to the reader early on in the paper. On a related note, Fig1b is difficult to parse.

**Strength And Weaknesses:**

The main strength of this paper is the finding that, for most OSR methods, 'false positive' predictions tend to be misclassified closed-set samples. Though this may seem obvious, I do not believe this has been reported in the OSR literature. The main weaknesses for me revolve around the way this finding is phrased, and some of the experimental setups.

Strengths:
* As mentioned above, it is interesting to know that false positive open-set predictions tend to be misclassified closed-set examples. This is useful to researchers as the authors highlight a specific sub-problem, separating InC from open-set samples, as a key challenge. It is also important to practitioners, as both open-set examples and incorrectly predicted samples should be rejected by a closed-set classifier.
* The paper provides a thorough set of experiments (though I have some questions about their execution and details, see weaknesses) to demonstrate interesting empirical findings, such as the one regarding outlier exposure mentioned above.
* As needed for an empirical paper, the authors provide extensive additional implementation details in the appendix. They further include code to reproduce the paper's empirical findings, which is greatly appreciated.

Weaknesses:
* I am not sure that framing UOSR as an entirely new task is warranted or helpful in this case (e.g with new evaluation datasets). Rather, it seems that this paper highlights that a specific subset of the OSR problem (delineating InC vs open-set) is the most interesting part of the problem, and a new evaluation metric which focusses just on this decision could be interesting.
* I am a little confused as to why authors have used non-standard datasets for evaluation. The authors rely heavily on a CIFAR100 (ID) vs TinyImageNet/LSUN (OoD) experiment to demonstrate their findings. This is more similar to an OoD evaluation rather than OSR. Standard practise in the OSR literature is to use held-out categories from a single dataset as 'unseen' (this is especially an issue for the LSUN experiment, whose taxonomy is unrelated to CIFAR100).
* I am not sure why the authors have used different methods to compare video and image datasets, rather than a unified evaluation. A number of the image-based methods could be equally run on video and vice-versa. This lessens the clarity of the findings.
* The authors compare and contrast training with Outlier Exposure to using pretrained models. However, I think it is important to distinguish the *data* from the *method* used. As such, I think a better experiment would be to pre-train models with the OE data and see if the findings hold. I would also suggest the authors discuss how much overlap there is in the semantics of the pre-training/outlier training set with the OoD test set.
* (Minor): The authors do not introduce AURC, which I understand is from the Selective Prediction literature. Given this paper is targeted at the OSR community, I would suggest explaining it.

**Summary Of The Paper:**

This paper tackles the problem of 'Unified Open-Set Recognition' (UOSR), a variant of open-set recognition (OSR) which also considers how to deal with misclassified closed-set test samples. Concretely, the closed-set test samples are split into InW and InC (wrongly and correctly classified samples respectively), and the model must predict 'unknown' (i.e have a high open-set score) for InW.

I would summarise the central finding of this paper as "False Positives in standard OSR tend to be misclassified closed-set samples" (by False Positives, I mean examples from the training classes which have been predicted as 'open-set', and by misclassified, I mean wrongly predicted within the closed-set classes).

Besides the above, the authors provide an empirical investigation into OSR and OoD techniques for the UOSR problem, presenting some interesting findings such as that Outlier Exposure helps the InC/open-set decision but hurts the InW/InC decision. They further tackle the UOSR problem in a few-shot manner, and present a combination of KNN and a softmax score to solve it.

**Summary Of The Review:**

Overall, I believe this paper has an interesting central finding for OSR which is intuitive but has, to my knowledge, not been explored in the OSR literature before. The paper further conducts relatively rigorous experiments and makes effort to provide all details and code. That said, I believe there are issues with the exact execution of some experiments and presentation of findings, which prevent the findings from being easily accessible to the reader.

UPDATE AFTER REBUTTAL:

I have now looked through the other reviews and the corresponding authors' responses. The authors have responded to my main concerns and clarified some of my confusions.

My fellow reviewers' main concern seems to be that Vaze et al. [1] already show a correlation between ID accuracy and OSR performance. Though it is, perhaps, a corollary that the 'False Positive' OSR predictions tend to be InW samples, I believe it is an important point which has not received sufficient emphasis in the literature. It further seems to me that the authors have provided substantial additional empirical evidence to respond to all reviewers' concerns. Thus I believe this paper provides a valuable empirical contribution to the field.

As such, I maintain my previous rating (8, Good Paper) and recommend the paper for acceptance.

[1] OSR: A Good Closed-Set Classifier is All You Need? Vaze et al, ICLR 2022

---

> ### Author Response · Authors · 2022-11-18
> **Response to Reviewer sZtz (5/5)**
>
> **[Q5]**: The authors do not introduce AURC, which I understand is from the Selective Prediction literature.
> **[A5]**: Thanks for your suggestion! We add the definition of AURC in Appendix E.
> ***
> **[Q6]**: The take home from the paper, that 'false positive' predictions tend to be misclassified closed-set samples, is not presented in a straightforward manner.
> **[A6]**: This is a very significant and valuable suggestion for our work, as we also find it difficult to summarize our finding into a straightforward expression. Many thanks! Your description 'False positive predictions in the OSR tend to be misclassified closed-set samples' is very refined. This is a perfect take-home message for the OSR researchers, but we believe that we should use 'InW samples have similar uncertainty distribution with OoD samples rather than InC samples' as it is a general expression which does not lean to UOSR or OSR.  We re-write the second paragraph of the Introduction in the modified manuscript. Feel free to check it to see our new description.
> ***
>
> Reference:
> [1] Wentao, et al. "Evidential deep learning for open set action recognition" (ICCV 2021)

---

> ### Author Response · Authors · 2022-11-18
> **Response to Reviewer sZtz (4/5)**
>
> **Table N** _UOSR benchmark in the image domain under the ResNet50 model. InD dataset is CIFAR100 while the OoD dataset is TinyImageNet. †, ‡, ♢ refer to OSR-based, SP-based, UOSR- based methods. OD: Outlier Data. N/G/R means No/Generated/Real OD. AUORC (%), AURC (×103) and Acc. (%) are reported. pt: pre-training._
> |                             |     |                 ||w/o pt||                  ||w/ pt||
> |:-----------------------------|:-----:|:--------:|:--------:|:--------:|:--------:|:--------:|:--------:|:--------:|:--------:|
> |                             |     |        |  UOSR   |         | OSR    |        |        UOSR     || OSR    |
> | Methods         | OD  |   ACC $\uparrow$  |  AURC $\downarrow$   | AUROC $\uparrow$  | AUROC $\uparrow$  | ACC $\uparrow$    | AURC $\downarrow$    | AUROC $\uparrow$  | AUROC $\uparrow$  |
> | ODIN†            | N   | 72.08  | 371.92  | 86.69  | 81.65  | 86.48  | 261.42  | 91.81  | 91.47  |
> | LC†              | N   | 72.08  | 372.55  | 84.16  | 76.37  | 86.48  | 280.69  | 87.68  | 85.16  |
> | OpenMax†         | N   | 73.64  | 361.22  | 85.33  | 76.23  | 86.43  | 261.53  | 90.29  | 85.86  |
> | MaxLogits†       | N   | 73.89  | 351.39  | 87.09  | 80.11  | 86.42  | 249.86  | 92.91  | 91.11  |
> | Entropy†         | N   | 73.89  | 355.54  | 86.25  | 78.14  | 86.43  | 257.36  | 91.21  | 87.22  |
> | OLTR†            | N   | 73.69  | 357.79  | 85.94  | 77.10  | 86.22  | 260.17  | 90.59  | 86.05  |
> | Ensemble†        | N   | 76.78  | 314.56  | 89.59  | 83.93  | 87.41  | 240.34  | 93.67  | 92.02  |
> | Vim†             | N   | 73.89  | 341.88  | 87.37  | 80.92  | 86.43  | 246.59  | 93.01  | 92.33  |
> | BCE‡             | N   | 73.29  | 369.91  | 84.29  | 74.59  | 86.39  | 257.74  | 90.79  | 86.50  |
> | TCP‡             | N   | 71.80  | 369.17  | 85.17  | 75.46  | 86.83  | 261.97  | 89.95  | 85.67  |
> | DOCTOR‡          | N   | 72.08  | 378.45  | 84.48  | 75.06  | 86.48  | 262.05  | 90.24  | 85.75  |
> | SIRC(MSP, z)♢    | N   | 73.13  | 358.93  | 85.77  | 76.67  | 86.44  | 260.08  | 90.53  | 85.98  |
> | SIRC(MSP, Res.）♢ | N   | 73.13  | 348.40  | 87.26  | 80.29  | 86.44  | 247.75  | 92.99  | 90.39  |
> | SIRC(-H, z）♢     | N   | 73.13  | 355.62  | 86.57  | 78.16  | 86.44  | 257.28  | 91.23  | 87.27  |
> | SIRC(-H, Res.）♢  | N   | 73.13  | 346.21  | 87.71  | 81.43  | 86.44  | 244.55  | 93.68  | 91.62  |
> | SoftMax †        | N   | 73.28  | 363.55  | 84.90  | 75.59  | 86.44  | 260.14  | 90.50  | 85.93  |
> | SoftMax (OE) †   | R   | 73.54  | 339.59  | 88.78  | 84.35  | __85.43__  | __255.77__  | __92.54__  | __90.65__  |
> | ARPL †           | N   | 73.03  | 345.84  | 88.13  | 80.49  | 84.67  | 301.27  | 87.01  | 83.49  |
> | ARPL+CS †        | R   | 72.78  | 349.50  | 87.65  | 82.81  | __83.60__  | __268.00__  | __92.00__  | __91.17__  |
> | MCD (Dropout ) †| N   | 76.49  | 375.01  | 82.25  | 79.21  | 87.21  | 301.71  | 83.57  | 81.69  |
> | MCD  †           | R   | 70.88  | 365.82  | 86.97  | 79.88  | __81.96__  | __287.21__  | __90.43__  | __86.59__  |
> | PROSER  †        | G   | 68.08  | 394.48  | 84.82  | 79.23  | 81.32  | 301.78  | 90.20  | 89.29  |
> | PROSER(EB) †     | R   | 71.82  | 366.65  | 86.06  | 81.95  | 85.06  | 269.79  | 90.84  | 90.38  |
> | VOS †            | G   | 73.44  | 356.68  | 86.65  | 79.72  | 86.62  | 249.24  | 92.94  | 91.37  |
> | VOS †            | R   | 73.18  | 331.12  | 89.96  | 85.78  | 85.93  | 251.44  | 92.92  | 91.34  |
> | OpenGAN †        | R   | 73.61  | 334.04  | 88.92  | 85.48  | 86.25  | 260.19  | 91.21  | 90.94  |

---

> ### Author Response · Authors · 2022-11-18
> **Response to Reviewer sZtz (3/5)**
>
> **[Q3]**: Why the authors used different methods to compare video and image datasets?
> **[A3]**: We follow the only OSR work in the video domain [1] to build our UOSR benchmark in the video domain, so the methods in the video domains are the same as [1]. There are some overlapping methods in the image and video domains, including SoftMax, Dropout, OpenMax, BCE, TCP, DOCTOR, and SIRC. The most important thing is that we want to show our finding is a general phenomenon across different datasets, settings and domains, and we believe our experiments are diverse enough to testify the conclusion.
> ***
>
> **[Q4]**: I think a better experiment would be to pre-train models with the OE data and see if the findings hold. I would also suggest the authors discuss how much overlap there is in the semantics of the pre-training/outlier training set with the OoD test set.
> **[A4]**: This is a very reasonable idea. In fact, we already have the experiment results that use both pre-training weights and OE data in the original manuscript.
>
> (1) In Table 5 and 6, the methods with outlier data including OE, ARPL+CS, MCD, EB, VOS, and OpenGAN have both the results w/o and w/ pre-training weights. The four left columns are the results w/o pre-training weights, and the four right columns are w/ pre-training weights. We put Table 5 here in Table N for your reference, and the results of using both outlier data and pre-training weights are __bold__. If we focus on SoftMax/OE, ARPL/ARPL+CS, and Dropout/MCD pairs, we can see that the OE is still helpful w/ pre-training, such as 92.54 for OE and 90.50 for SoftMax.
>
> (2) In Fig. 4 of the manuscript, we also provide the results using both pre-training weights and OE data, which are marked as TP w/ OE (TP means train from pre-training weights) and in <font color=green>green color</font>. Comparing TP w/ OE with TP (<font color=green>green marks</font> v.s.<font color=brwon>brown marks</font>), we can see the InC/InW discrimination is lower and InC/OoD discrimination is higher, which is consistent with the conclusion without pre-training.
>
> (3) The overlap of outlier training set with the OoD test set is already discussed in 'Outlier datasets' parapraph in Appendix E.2 and E.3. The pre-training datasets for both image and video domains are so large that they contain the overlapping classes with test OoD dataset.

---

> ### Author Response · Authors · 2022-11-18
> **Response to Reviewer sZtz (2/5)**
>
> **[Q2]**: Why use non-standard OSR datasets for evaluation?
> **[A2]**: (1) The reason we use one dataset as InD and another dataset as OoD is that we want to unify the dataset setting for image and video domains. The only OSR work in the video domain [1] uses this setting, so we follow them and use this setting in the video domain. Then we naturally use this setting in the image domain so that we use the same setting for image and video domains. Considering that the only difference between OSR and OoD detection is the dataset configuration, and their goals are the same to accept InD and reject OoD samples, we think this dataset setting is acceptable.
>
> (2) We provide the results of traditional OSR settings on the most challenging TinyImageNet in Table M. We can see that our finding still holds (UOSR has better performance than OSR for the same method, and InW/OoD discrimination is much worse than InC/OoD). Surprisingly, we find the InW/OoD AUORC of ODIN method achieves 84.32, which means InW and OoD samples can be well distinguishable. This proves our claim in Appendix D that the features of InC/InW/OoD samples are separable and it is possible to find a proper uncertainty estimation method to distinguish these three groups of data. Please see Appendix D for more details.
>
> **Table M** _UOSR benchmark in the image domain under the traditional OSR dataset setting. All methods are conducted under the R50 model. Dataset is TinyImageNet. †, ‡, ♢ refer to OSR-based, SP-based, UOSR-based methods. Pre-training weights are used._
> |                  |        | UOSR    |        | OSR    |InC/InW |InC/OoD |InW/OoD |
> |:------------------|:--------:|:--------------:|:-------------:|:-------------:|:-------:|:-------------:|:-------------:|
> | Methods          | Acc. $\uparrow$  | AURC $\downarrow$        | AUROC $\uparrow$      | AUROC $\uparrow$      | AUROC $\uparrow$| AUROC $\uparrow$      | AUROC $\uparrow$      |
> | SoftMax          | 87.40  | 730.10  | 94.06  | 90.59  | 91.02  | 94.11  | 66.21  |
> | ODIN             | 87.40  | 724.90  | 96.62  | 95.21  | 85.50  | 96.78  | 84.32  |
> | LC               | 87.40  | 756.49  | 88.02  | 84.32  | 81.27  | 88.11  | 58.05  |
> | OpenMax          | 86.60  | 731.73  | 94.49  | 90.25  | 91.77  | 94.53  | 62.59  |
> | OLTR             | 87.60  | 731.93  | 93.69  | 90.09  | 89.96  | 93.74  | 64.34  |
> | PROSER           | 86.90 |750.26 |92.29 |90.21 |77.01 |92.51 |74.91  |
> | BCE              | 87.60  | 732.79  | 93.74  | 90.46  | 89.63  | 93.80  | 66.92  |
> | TCP              | 87.70  | 731.66  | 94.20  | 90.54  | 90.20  | 94.25  | 64.06  |
> | DOCTOR           | 87.40  | 731.69  | 93.97  | 90.54  | 90.25  | 94.02  | 66.40  |
> | SIRC(MSP, z）     | 87.40  | 731.48  | 94.02  | 90.53  | 91.04  | 94.06  | 66.06  |
> | SIRC(MSP, Res.)  | 87.40  | 729.96  | 94.77  | 91.71  | 91.19  | 94.82  | 70.13  |
> | SIRC(-H, z）      | 87.40  | 730.12  | 94.74  | 91.65  | 91.24  | 94.79  | 69.89  |
> | SIRC(-H, Res.)   | 87.40  | 731.34  | 94.06  | 90.59  | 91.02  | 94.10  | 66.22  |

---

> ### Author Response · Authors · 2022-11-18
> **Response to Reviewer sZtz (1/5)**
>
> We are so glad to see that the reviewer sZtz acknowledges the importance of our finding that InW samples have similar uncertainty distribution with OoD samples rather than InC samples, or false positive predictions for most OSR methods are actually InW samples. This is also the point that we want to let more researchers know, so that they can take it into account when designing new methods. The answers to your feedback are provided below. We also give more insights about our finding in Appendix D which we analyze our finding from the feature and uncertainty score estimation perspectives. The summary of modifications in our manuscript is in the 'Manuscript modification summary' public comment.
> ***
> **[Q1]**: I am not sure that framing UOSR as an entirely new task is warranted or helpful in this case, as we care about a subset of OSR problem.
> **[A1]**: We understand your claim that we can regard UOSR as a sub-task of OSR and use other additional metrics to evaluate the UOSR performance, but we believe it is better to regard UOSR as a new task for these reasons:
>
> (1) The UOSR and OSR are contradictory to some extent so it may not be suitable to put them into one task. UOSR aims to reject InW samples while OSR aims to accept them. If there are several sub-tasks within one task, the goals of sub-tasks should not be contradictory with each other. For example, better bounding box classification and bounding box size estimation in object detection are not contradictory, and both of them aim to detect objects better. Therefore, to avoid confusing future researchers, we think regarding UOSR as a new task is more suitable.
>
> (2) We think the UOSR has great
> value in both practical and academic perspectives, so it is significant enough to be regarded as a new task. In real-world applications, both the evaluation results of InW samples and OoD samples need to be corrected, so we should not use them. For example, in the autonomous driving scenario, most OSR works said we should reject OoD samples such as a golf car so that the autonomous vehicle will not crash on it. However, if the model classifies a normal car as the drivable road during semantic segmentation, it is still dangerous. In most cases InW samples should be rejected, like selective prediction (SP) expects (accept InC and reject InW samples), so UOSR may be more practical than OSR in many real-world applications. As for the academic perspective, we already show in our manuscript that most existing OSR works actually perform better under the UOSR scenario, which provides good baselines for future research. We also notice that research in SP is very rare compared to OSR, but SP is a significant part of UOSR. So we hope the research of UOSR and SP can boost each other as both of them aim to disinguish InC and InW samples. Our proposed FS-KNNS for the few-shot UOSR is actually a good example, as we show how to improve InC/OoD discrimination without losing InC/InW performance.

---

### Official Review · Reviewer_ZHBz · 2022-10-23

**Confidence:** 4
**Correctness:** 3
**Technical Novelty And Significance:** 2
**Empirical Novelty And Significance:** 3
**Recommendation:** 6

**Clarity, Quality, Novelty And Reproducibility:**

- The language of the article is clear and the overall quality is good.
- Although the article does not propose a novel model in terms of method, it raises the problem of existing OSR, which will provide guidance for future work, and the article is relatively novel.
- The authors provide the source code as supplementary material and the results are reproducible.

**Strength And Weaknesses:**

Strength:
- This article deeply analyzes the UOSR task under different training and evaluation settings through extensive experiments.
- Although Vaze et al. [7] have demonstrated the importance of a good closed set, this paper further illustrates the impact of InW samples on OSR performance, providing a meaningful reference for future OOD detection research.
- The article analyzes how pre-training and outlier exposure influence the UOSR.
- This paper proposed a new evaluation setting called few-shot UOSR, and proposed a new method named FS-KNNS.
- Detailed ablation experiments are included in the appendix of this paper, and the results of the paper are reproducible.

Weaknesses:
- In Section 2, the authors mentioned the definition of UOSR, which rejects InW and OoD samples. The authors may also consider discussing outlier detection [1,2] as related work.
- Some OSR methods need to use the generated OoD samples to assist in training [3,8], while some OSR methods only need to train InD samples to identify OoD samples [4]. This article should discuss whether the InW samples will play an equally important role in the above cases in Tables 2 and 3. Perhaps it is possible to report the performance of OSR methods with/without OoD sample-assisted training separately in the table.
- The authors discuss the impact of InW on UOSR. Did the authors consider the effect of some noisy outliers in the training set on the results? Or just consider the impact of misclassified samples in the test set on the results? How to discover them and use them to improve the model? I am glad to see related discussions.
- In Table 5, the authors compare the methods of outlier exposure and claim that the method of outlier exposure can effectively improve UOSR. However, I am concerned that this is the superiority of the OOD detection method itself, since the methods not based on outlier exposure are older. Should the author try to introduce outlier exposure settings for the same method and observe the AUROC changes to confirm the final effect, or try to compare some SOTA methods that do not require outlier exposure, e.g., [5,6]?

Some Questions:
- Appendix C mentioned the outlier datasets used for the study. However, the outliers of other methods appear to be generated from known data [3,8], while the authors use additional data directly. Should the authors discuss the difference between using real and virtual outliers, and how they relate to the conclusions drawn in this paper?
- The InD dataset used for OSR in this paper is CIFAR-100, a relatively simple dataset. Whether the author considers using InD dataset with large-scale semantic space such as TinyImageNet, because this dataset seems to be more challenging and more realistic to reflect the negative effects of InW, I am glad to see related discussions.

Overall:
This paper mentions many important issues that have not been paid attention to by the current research on OoD detection. However, there are still some problems with this article. If the authors can provide convincing replies to the above queries, I am willing to improve my initial score.


[1] Goodge, Adam, et al. "Lunar: Unifying local outlier detection methods via graph neural networks." AAAI. 2022.
[2] Chauhan, Kushal, et al. "Robust outlier detection by de-biasing VAE likelihoods." CVPR. 2022.
[3] Wang, Yezhen, et al. "Energy-based open-world uncertainty modeling for confidence calibration." CVPR. 2021.
[4] Bao, Wentao, Qi Yu, and Yu Kong. "Evidential deep learning for open set action recognition." ICCV. 2021.
[5] Cao, Senqi, and Zhongfei Zhang. "Deep Hybrid Models for Out-of-Distribution Detection." CVPR. 2022.
[6] Wang, Haoqi, et al. "ViM: Out-Of-Distribution with Virtual-logit Matching." CVPR. 2022.
[7] Vaze, Sagar, et al. "Open-Set Recognition: A Good Closed-Set Classifier is All You Need." ICLR. 2021.
[8] Du, Xuefeng, et al. "VOS: Learning What You Don't Know by Virtual Outlier Synthesis." ICLR. 2021.





**Summary Of The Paper:**

This article demonstrates the uncertainty distribution of almost all existing OSR methods is actually closer to the expectation of UOSR than OSR. This article also introduces a new evaluation setting into UOSR, which is few-shot UOSR. Then, the FS-KNNS method is proposed, which achieves state-of-the-art performance under all settings.

**Summary Of The Review:**

This paper mentions many important issues that have not been paid attention to by the current research on OoD detection. However, there are still some problems with this article. If the authors can provide convincing replies to the above issues, I am willing to improve my initial score.

---

> ### Author Response · Authors · 2022-11-18
> **Response to Reviewer ZHBz (6/6)**
>
> **[Q5]**: Discussion about the difference between using generated outlier data and real outlier data.
> **[A5]**: This is also a very interesting ablation experiment. We include two methods, i.e., PROSER and VOS, into the Table I when they use generated outlier data and real outlier data. PROSER uses mixup samples as generated data and VOS uses samples at the edge of InD sample distributions as generated data. From Table I we can see that using real outlier data has better performance than using generated outlier data in general. The reason is that InC/OoD discrimination is better when using real outlier data (88.36-87.04 for PROSER; 91.44-87.51 for VOS in Table J). We think real outlier data distribution is more similar with OoD samples in the test set, so using real outlier data as the proxy of OoD samples are more suitable.
> ***
> **[Q6]**: The results of using InD dataset with large-scale semantic space such as TinyImageNet.
> **[A6]**: Thanks a lot for this kind suggestion to enrich our work. We provide the UOSR evaluation results in Table L when InD and OoD datasets are TinyImageNet and CIFAR100 respectively. Pre-training weights are used. TinyImageNet has 200 classes in the training set which is more diverse than CIFAR100. We can see that training the model on TinyImageNet is more challenging than CIFAR100, as the Acc. of CIFAR100 is 86.44 and Acc. of TinyImageNet only reaches 77.02. In this way, the impact of InW samples becomes further huge. For example, the performance gap between UOSR and OSR for the SoftMax method is 8.95 when InD dataset is TinyImageNet, and this value is only 0.34 when InD dataset is CIFAR100. So the InW samples are more important for the performance when InD dataset is difficult, as lower closed-set Acc. means more InW samples. Our findings that InW samples share similar uncertainty distribution with OoD samples still holds, as the InW/OoD AUROC is much lower than InC/InW.
>
> **Table L** _UOSR benchmark in the image domain. All methods are conducted under the R50 model. InD and OoD Dataset are TinyImageNet and CIFAR100 respectively. †, ‡, ♢ refer to OSR-based, SP-based, UOSR-based methods. Pre-training weights are used._
> |                  |        | UOSR    |        | OSR    |InC/InW |InC/OoD |InW/OoD |
> |:------------------|:--------:|:--------------:|:-------------:|:-------------:|:-------:|:-------------:|:-------------:|
> | Methods          | Acc. $\uparrow$  | AURC $\downarrow$        | AUROC $\uparrow$      | AUROC $\uparrow$      | AUROC $\uparrow$| AUROC $\uparrow$      | AUROC $\uparrow$      |
> | SoftMax †         | 77.02  | 340.70  | 86.22  | 77.27  | 88.83  | 85.62  | 49.27  |
> | ODIN †            | 77.23  | 359.57  | 84.01  | 75.32  | 86.53  | 83.43  | 47.81  |
> | LC †              | 77.23  | 385.15  | 79.76  | 71.85  | 83.21  | 78.98  | 47.69  |
> | OpenMax †         | 76.90  | 340.21  | 86.53  | 77.09  | 89.63  | 85.81  | 48.04  |
> | OLTR †            | 77.00  | 341.60  | 86.04  | 76.75  | 89.02  | 85.36  | 47.92  |
> | PROSER †          | 75.93  | 392.91  | 80.50  | 74.65  | 79.20  | 80.50  | 55.20  |
> | BCE ‡             | 76.91  | 339.43  | 86.40  | 76.83  | 89.64  | 85.66  | 47.44  |
> | TCP  ‡            | 77.82  | 336.61  | 86.48  | 77.47  | 89.72  | 85.76  | 48.38  |
> | DOCTOR  ‡         | 77.23  | 339.02  | 86.69  | 77.62  | 89.86  | 85.96  | 49.30  |
> | SIRC(MSP, z）♢     | 77.03  | 337.28  | 86.82  | 78.86  | 88.78  | 86.37  | 53.65  |
> | SIRC(MSP, Res.) ♢ | 77.03  | 316.14  | 90.66  | 87.00  | 88.82  | 91.08  | 73.31  |
> | SIRC(-H, z）♢      | 77.03  | 333.73  | 87.60  | 80.13  | 89.00  | 87.28  | 56.19  |
> | SIRC(-H, Res.) ♢  | 77.03  | 311.98  | 91.39  | 88.20  | 89.02  | 91.94  | 75.67  |
> ***
>
> Reference:
> [1] Adam, et al. "LUNAR: Unifying Local Outlier Detection Methods via Graph Neural Networks" (AAAI 2022)
> [2] Kushal, et al. "Robust outlier detection by de-biasing VAE likelihoods" (CVPR 2022)
> [3] Lakshminarayanan, et al. "Simple and scalable predictive uncertainty estimation using deep ensembles" (NeurIPS 2017)
> [4] Hendrycks, et al. "Scaling Out-of-Distribution Detection for Real-World Settings" (ICML 2022)
> [5] Haoqi, et al. "ViM: Out-Of-Distribution with Virtual-logit Matching" (CVPR 2022)
> [6] Guangyao, et al. "Adversarial Reciprocal Points Learning for Open Set Recognition" (TPAMI 2022)
> [7] Kong, et al. "OpenGAN: Open-Set Recognition via Open Data Generation" (ICCV 2021)
> [8] Zhifan, et al. "NGC: A Unified Framework for Learning with Open-World Noisy Data" (ICCV 2021)
> [9] Senqi, et al. "Deep Hybrid Models for Out-of-Distribution Detection" (CVPR 2022)
> [10] Qing, et al. "Unsupervised out-of-distribution detection by maximum classifier discrepancy" (ICCV 2019)
> [11] Sunil, et al. "An Effective Baseline for Robustness to Distributional Shift" (ICMLA 2021)

---

> ### Author Response · Authors · 2022-11-18
> **Response to Reviewer ZHBz (5/6)**
>
> **[Q4]**: I am concerned that the superiority of outlier data comes from the OOD detection method itself, since the methods not based on outlier exposure are older. Should the author try to introduce outlier exposure settings for the same method and observe the AUROC changes to confirm the final effect, or try to compare some SOTA methods that do not require outlier exposure, e.g., [5,9]?
> **[A4]**: Thanks for this very general reminder. (1) When researchers want to use the outlier data for better OSR performance, they naturally design a totally new method to make full use of outlier data, like MCD[10] and EB[11]. So the method designed w/ outlier data is different from the method w/o outlier data, and there is no perfectly fair comparison between two methods w/o and w/ outlier data. In this way, we try our best to compare two methods w/o and w/ outlier data whose methods are as similar as possible, including SoftMax/OE, ARPL/ARPL+CS, and Dropout/MCD in Table I. We can see for most cases, outlier data is helpful for the UOSR and OSR performance. One exception is ARPL has better UOSR performance than ARPL+CS w/o pre-training. The reason is that although ARPL+CS has better InC/OoD discrimination than ARPL (89.36-88.76 in Table J), the InC/InW discrimination is worse (81.40-85.77), which is consistent with our conclusion that outlier data is beneficial for InC/OoD performance but may deteriorate the InC/InW performance. So outlier data may not improve the overall UOSR performance (SoftMax has better UOSR performance than EB/VOS in the video domain (88.42-86.10/87.98 in Table 6 in the manuscript)). Please see the 'Analysis' Paragraph in Sec. 4 for more details.
>
> (2) We introduce several additional OSR methods w/o outlier exposure into the benchmark, including Ensemble[3], MaxLogits[4], Entropy, Vim[5], ARPL[6], ARPL+CS[6] and OpenGAN[7]. [9] is not included as they do not open their code. We can see that Ensemble has better UOSR performance than almost all methods w/ outlier data in Table I, although it needs to inference the model 5 times which consumes much more time. So we think your concern is very reasonable. We should compare two methods w/ and w/o outlier data when their methods are very similar, like SoftMax/OE, ARPL/ARPL+CS, and Dropout/MCD in Table I, so that we can see whether outlier data really helps or not.

---

> ### Author Response · Authors · 2022-11-18
> **Response to Reviewer ZHBz (4/6)**
>
> **[Q3]**: The effect of some noisy outliers in the training set on the results. How to discover them and use them to improve the model.
> **[A3]**: This is a very interesting extension of our work. We change the labels of some outlier data to the labels of InD classes, and study how these noisy outlier data influence the performance. In addition, we use NGC[8] to find the noisy outlier data and correct them. The results are shown in Table K. We can see that the closed-set Acc. gradually decreases with the growth of noise level when NGC is not used. This is natural as some noisy outlier data corrupt the InD data distribution. In contrast, the closed-set Acc. does not drop with the growth of noise level when NGC is used. So NGC is very effective in finding corrupted samples. When the noise level is 0\%, the UOSR performance with NGC is better than the performance without NGC (lower AURC value), meaning that other parts in NGC that are not related to label corruption, such as contrastive learning, are helpful for UOSR. Surprisingly, the performance of UOSR and OSR are relatively stable under different noise levels no matter we use NGC or not, compared to the clear performance drop of closed-set Acc. when NGC is not used. So the model is robust in open-set related performance when noisy outlier data is introduced. Our finding that InD samples share similar uncertainty scores with OoD samples still holds as AUROC of InW/OoD is close to 50.
>
> **Table K** _UOSR OSR performance under noisy outlier data. InD dataset is CIFAR100 and outlier dataset is 300K Random Images. OoD dataset is TinyImageNet. Experiments are conducted with ResNet18 backbone._
> |             |     |        |        UOSR     || OSR    |InC/InW |InC/OoD |InW/OoD |
> |-------------|-----|--------|---------|--------|--------|--------|--------|--------|
> | Noise Level | NGC | Acc. $\uparrow$  | AURC $\downarrow$   | AUROC $\uparrow$ | AUROC $\uparrow$ | AUROC $\uparrow$ | AUROC $\uparrow$ | AUROC $\uparrow$ |
> | 0%          | ×   | 76.19  | 334.13  | 85.50  | 77.29  | 85.05  | 85.60  | 50.70  |
> | 20%         | ×   | 72.15  | 343.46  | 87.84  | 77.53  | 87.34  | 87.98  | 50.45  |
> | 40%         | ×   | 70.06  | 343.60  | 89.91  | 79.86  | 88.36  | 90.37  | 55.25  |
> | 60%         | ×   | 69.46  | 346.62  | 90.07  | 79.79  | 88.76  | 90.47  | 55.52  |
> | 80%         | ×   | 69.20  | 348.18  | 89.95  | 79.05  | 88.84  | 90.29  | 53.77  |
> | 100%        | ×   | 68.05  | 356.35  | 89.97  | 78.31  | 89.63  | 90.08  | 53.23  |
> | 0%          | ✓   | 77.52  | 315.32  | 87.75  | 81.12  | 85.61  | 88.23  | 56.61  |
> | 20%         | ✓   | 77.50  | 316.86  | 87.81  | 80.46  | 86.88  | 88.02  | 54.40  |
> | 40%         | ✓   | 76.87  | 321.66  | 87.58  | 79.48  | 87.15  | 87.68  | 52.23  |
> | 60%         | ✓   | 77.07  | 319.58  | 87.91  | 79.88  | 87.43  | 88.02  | 52.53  |
> | 80%         | ✓   | 77.12  | 317.90  | 88.17  | 80.14  | 87.41  | 88.34  | 52.49  |
> | 100%        | ✓   | 77.38  | 328.85  | 86.24  | 78.00  | 86.46  | 86.20  | 49.94  |

---

> ### Author Response · Authors · 2022-11-18
> **Response to Reviewer ZHBz (3/6)**
>
> **Table J** _Uncertainty distribution analysis in image domain with ResNet50. Pre-training is not used. InD dataset: CIFAR100. OoD dataset: TinyImageNet. AUORC (%) is reported. N/G/R means No/Generated/Real Outlier data._
> | Methods          | OD   | InC/OoD   | InC/InW  | InW/OoD  | OSR  | UOSR  |
> |:------------------|:--------:|:---------:|:--------:|:--------:|:--------:|:--------:|
> | ODIN |N |88.35 |80.76 |64.36 |81.65 |86.69
> | LC |N |84.60 |82.58 |55.14 |76.37 |84.16|
> | OpenMax |N |85.16 |85.96 |51.27 |76.23 |85.33
> | OLTR |N |85.99 |85.74 |52.22 |77.10 |85.94
> | SoftMax |N  |84.69 |85.68 |50.64 |75.59 |84.90
> |SoftMax(OE) |R  |90.04 |84.00 |68.54 |84.35 |88.78|
> |ARPL |N  |88.76 |85.77 |58.09 |80.49 |88.13|
> |ARPL+CS |R  |89.36 |81.40 |65.32 |82.81 |87.65|
> |MCD(Dropout) |N  |84.15 |74.17 |63.15 |79.21 |82.25|
> |MCD |R  |87.47 |85.23 |61.39 |79.88 |86.97|
> |PROSER |G |87.04 |77.84 |62.57 |79.23 |84.82|
> |PROSER (EB) |R  |88.36 |77.88 |65.60 |81.95 |86.06|
> |VOS |G |87.51 |83.38 |58.17 |79.72 |86.65 |
> |VOS |R  |91.44 |84.41 |70.32 |85.78 |89.96|

---

> ### Author Response · Authors · 2022-11-18
> **Response to Reviewer ZHBz (2/6)**
>
> **[Q2]**: Whether the InW samples will play an equally important role when generated OoD samples are used or not used during training.
> **[A2]**: This is a very nice suggestion. We include several additional methods into the UOSR benchmark, including Ensemble[3], MaxLogits[4], Entropy, Vim[5], ARPL[6], ARPL+CS[6] and OpenGAN[7]. We then reorganize the benchmark according to your concerns Q2, Q4, and Q5. The final results are in Table I. We can see that for two methods using generated outlier data (PROSER and VOS), the UOSR performance is still better than OSR w/o and w/ pre-training. Then we provide a more detailed analysis in Table J. We can see that InC/OoD and InC/InW discrimination of PROSER and VOS are significantly better than InW/OoD like other methods w/o generated outlier data, which illustrates that InW samples also share similar uncertainty distribution with OoD samples rather than InC samples when generated outlier data is introduced.
>
> **Table I** _UOSR benchmark in the image domain under the ResNet50 model. InD dataset is CIFAR100 while the OoD dataset is TinyImageNet. †, ‡, ♢ refer to OSR-based, SP-based, UOSR- based methods. OD: Outlier Data. N/G/R means No/Generated/Real OD. AUORC (%), AURC (×103) and Acc. (%) are reported. pt: pre-training._
> |                             |     |                 ||w/o pt||                  ||w/ pt||
> |:-----------------------------|:-----:|:--------:|:--------:|:--------:|:--------:|:--------:|:--------:|:--------:|:--------:|
> |                             |     |        |  UOSR   |         | OSR    |        |        UOSR     || OSR    |
> | Methods         | OD  |   ACC $\uparrow$  |  AURC $\downarrow$   | AUROC $\uparrow$  | AUROC $\uparrow$  | ACC $\uparrow$    | AURC $\downarrow$    | AUROC $\uparrow$  | AUROC $\uparrow$  |
> | ODIN†            | N   | 72.08  | 371.92  | 86.69  | 81.65  | 86.48  | 261.42  | 91.81  | 91.47  |
> | LC†              | N   | 72.08  | 372.55  | 84.16  | 76.37  | 86.48  | 280.69  | 87.68  | 85.16  |
> | OpenMax†         | N   | 73.64  | 361.22  | 85.33  | 76.23  | 86.43  | 261.53  | 90.29  | 85.86  |
> | MaxLogits†       | N   | 73.89  | 351.39  | 87.09  | 80.11  | 86.42  | 249.86  | 92.91  | 91.11  |
> | Entropy†         | N   | 73.89  | 355.54  | 86.25  | 78.14  | 86.43  | 257.36  | 91.21  | 87.22  |
> | OLTR†            | N   | 73.69  | 357.79  | 85.94  | 77.10  | 86.22  | 260.17  | 90.59  | 86.05  |
> | Ensemble†        | N   | 76.78  | 314.56  | 89.59  | 83.93  | 87.41  | 240.34  | 93.67  | 92.02  |
> | Vim†             | N   | 73.89  | 341.88  | 87.37  | 80.92  | 86.43  | 246.59  | 93.01  | 92.33  |
> | BCE‡             | N   | 73.29  | 369.91  | 84.29  | 74.59  | 86.39  | 257.74  | 90.79  | 86.50  |
> | TCP‡             | N   | 71.80  | 369.17  | 85.17  | 75.46  | 86.83  | 261.97  | 89.95  | 85.67  |
> | DOCTOR‡          | N   | 72.08  | 378.45  | 84.48  | 75.06  | 86.48  | 262.05  | 90.24  | 85.75  |
> | SIRC(MSP, z)♢    | N   | 73.13  | 358.93  | 85.77  | 76.67  | 86.44  | 260.08  | 90.53  | 85.98  |
> | SIRC(MSP, Res.）♢ | N   | 73.13  | 348.40  | 87.26  | 80.29  | 86.44  | 247.75  | 92.99  | 90.39  |
> | SIRC(-H, z）♢     | N   | 73.13  | 355.62  | 86.57  | 78.16  | 86.44  | 257.28  | 91.23  | 87.27  |
> | SIRC(-H, Res.）♢  | N   | 73.13  | 346.21  | 87.71  | 81.43  | 86.44  | 244.55  | 93.68  | 91.62  |
> | __SoftMax †__        | __N__   | __73.28__  | __363.55__  | __84.90__  | __75.59__  | __86.44__  | __260.14__  | __90.50__  | __85.93__  |
> | __SoftMax (OE) †__   | __R__   | __73.54__  | __339.59__  | __88.78__  | __84.35__  | __85.43__  | __255.77__  | __92.54__  | __90.65__  |
> | __ARPL †__           | __N__   | __73.03__  | __345.84__  | __88.13__  | __80.49__  | __84.67__  | __301.27__  | __87.01__  | __83.49__  |
> | __ARPL+CS †__        | __R__   | __72.78__  | __349.50__  | __87.65__  | __82.81__  | __83.60__  | __268.00__  | __92.00__  | __91.17__  |
> | __MCD (Dropout ) †__| __N__   | __76.49__  | __375.01__  | __82.25__  | __79.21__  | __87.21__  | __301.71__  | __83.57__  | __81.69__  |
> | __MCD  †__           | __R__   | __70.88__  | __365.82__  | __86.97__  | __79.88__  | __81.96__  | __287.21__  | __90.43__  | __86.59__  |
> | __PROSER  †__        | __G__   | __68.08__  | __394.48__  | __84.82__  | __79.23__  | __81.32__  | __301.78__  | __90.20__  | __89.29__  |
> | __PROSER(EB) †__     | __R__   | __71.82__  | __366.65__  | __86.06__  | __81.95__  | __85.06__  | __269.79__  | __90.84__  | __90.38__  |
> | __VOS †__            | __G__   | __73.44__  | __356.68__  | __86.65__  | __79.72__  | __86.62__  | __249.24__  | __92.94__  | __91.37__  |
> | __VOS †__            | __R__   | __73.18__  | __331.12__  | __89.96__  | __85.78__  | __85.93__  | __251.44__  | __92.92__  | __91.34__  |
> | OpenGAN †        | R   | 73.61  | 334.04  | 88.92  | 85.48  | 86.25  | 260.19  | 91.21  | 90.94  |

---

> ### Author Response · Authors · 2022-11-18
> **Response to Reviewer ZHBz (1/6)**
>
> We thank the reviewer for giving several constructive suggestions that can effectively improve the clarification of our work, especially about the outlier data part. All concerns are well handled below, and we also have some new insights and analysis in the modified manuscript (red paragraphs), such as why InW samples share similar uncertainty with OoD samples instead of InC samples from the feature and uncertainty estimation perspective in Appendix D. The summary of modification in our manuscript is in the 'Manuscript modification summary' public comment.
> ***
> **[Q1]**: The authors may also consider discussing outlier detection [1,2] as related work.
> **[A1]**: Thanks for the valuable advice which makes our work more complete. We already add them to our related work section. We discuss Anomaly detection and Outlier detection together as both of them do not require InD classification.

---

### Official Review · Reviewer_zcng · 2022-10-24

**Confidence:** 4
**Correctness:** 3
**Technical Novelty And Significance:** 2
**Empirical Novelty And Significance:** 3
**Recommendation:** 6

**Clarity, Quality, Novelty And Reproducibility:**

- Paper is generally well-written and easy to follow.
- Novelty is somewhat limited as discussed above.
- Code is provided in the supplementary, though I didn't get a chance to run to verify.

**Strength And Weaknesses:**

Strength
- The paper introduced a relatively thorough analysis of OSR methods for UOSR, revealing that the wrongly classified samples from known classes have a huge impact on the performance.
- The paper is generally well-written and easy to follow.

Weaknesses
- Though a good set of analysis is given, some of them appear to be straightforward and intuitive. For example, The wrongly classified samples from known classes have a strong impact on the final performance, which appears to be very obvious especially when the correlation between InD performance and OoD performance is founded (Vaze et al, 2022) because better  InD performance means less InW and more InC.  Hence, the additional value of the study here seems to be limited.
- It is unclear why state-of-art methods such as ARPL, ARPL+CS, and OpenGAN are missed in the comparison and analysis.
- The SSB benchmark introduced by Vaze et al, 2022 appears to be better suited for OSR performance evaluation (thus UOSR), especially when there are fine-grained datasets and ImageNet-scale OSR data. It would be more convincing to evaluate the method on SSB.

**Summary Of The Paper:**

This paper addresses an extended problem of open-set recognition (OSR), called Unified Open-set Recognition (UOSR). Unlike OSR, which only rejects testing samples from classes that the model has not seen during training, UOSR aims at rejecting both samples from unseen classes and also samples from seen classes but wrongly classified. The paper aims at providing a comprehensive study of the UOSR problem by adopting existing OSR methods. Meanwhile, analysis is also given on the effects of supervised pretraining on ImageNet or Kinetics 400 and the effects of training the model with outlier exposure with extra out-of-distribution samples. Finally, the paper introduces a simple technique for few-shot UOSR by introducing a KNN-based margin in SoftMax for uncertainty estimation, showing improved performance compared with the baselines.

**Summary Of The Review:**

Overall, I find this paper interesting with promising analysis. However, given the concerns above on the significance of the findings, which appear to be quite straightforward after having the findings by Vaze et al 2022, as well as the concerns on compared methods and datasets, I would hold a relatively conservative view about the paper and not recommend for acceptance.

---

> ### Author Response · Authors · 2022-11-18
> **Response to Reviewer zcng (4/4)**
>
> **Table G** _UOSR in the image domain. All methods are conducted under the R50 model. Dataset is Fine-Grained Visual Classification of Aircraft (FGVC-Aircraft). †, ‡, ♢ refer to OSR-based, SP-based, UOSR-based methods. Pre-training weights are used. EASY/HARD modes are reported._
> |                  |        |     UOSR      |             | OSR         |InC/InW |InC/OoD      |InW/OoD      |
> |:------------------|:--------:|:--------------:|:-------------:|:-------------:|:-------:|:-------------:|:-------------:|
> | Methods          | Acc. $\uparrow$  | AURC $\downarrow$        | AUROC $\uparrow$      | AUROC $\uparrow$      | AUROC $\uparrow$| AUROC $\uparrow$      | AUROC $\uparrow$      |
> | SoftMax †         | 85.61  | 129.71/127.18 | 89.46/82.46 | 84.24/72.85 | 88.54  | 89.80/79.08 | 51.18/35.75 |
> | ODIN †            | 85.25  | 152.17/173.15 | 87.11/75.75 | 84.92/68.36 | 80.04  | 89.72/73.32 | 57.17/39.70 |
> | LC †              | 85.25  | 134.44/144.29 | 88.98/80.15 | 83.55/69.74 | 87.58  | 89.50/75.93 | 49.13/33.95 |
> | OpenMax †         | 86.69  | 123.31/123.86 | 90.13/82.10 | 85.63/73.20 | 88.12  | 90.80/79.01 | 51.99/35.36 |
> | OLTR †            | 85.97  | 128.01/124.92 | 89.86/82.79 | 85.11/73.45 | 88.57  | 90.31/79.66 | 53.26/35.35 |
> | PROSER †          | 85.97  | 124.15/137.70 | 90.73/80.40 | 86.92/70.44 | 87.31  | 91.93/76.67 | 56.24/32.22 |
> | BCE ‡             | 85.37  | 134.56/127.05 | 88.75/82.37 | 84.05/73.93 | 86.49  | 89.58/80.06 | 51.75/38.17 |
> | TCP  ‡            | 85.25  | 132.09/131.80 | 89.30/82.19 | 83.60/72.12 | 88.41  | 89.63/78.66 | 48.73/34.30 |
> | DOCTOR ‡          | 85.25  | 133.93/144.07 | 89.16/80.24 | 83.79/69.80 | 87.68  | 89.71/76.00 | 49.58/33.96 |
> | SIRC(MSP, z）♢     | 85.61  | 130.50/126.36 | 89.24/82.57 | 83.86/73.04 | 88.52  | 89.49/79.27 | 50.31/35.98 |
> | SIRC(MSP, Res.) ♢ | 85.61  | 129.71/127.18 | 89.46/82.46 | 84.24/72.85 | 88.54  | 89.80/79.08 | 51.17/35.76 |
> | SIRC(-H, z）♢      | 85.61  | 128.97/125.59 | 89.71/82.85 | 84.70/73.60 | 88.55  | 90.13/79.70 | 52.40/37.35 |
> | SIRC(-H, Res.) ♢  | 85.61  | 128.04/126.58 | 89.97/82.69 | 85.16/73.35 | 88.52  | 90.49/79.46 | 53.46/37.01 |
>
> **Table H** _UOSR benchmark in the image domain. All methods are conducted under the R50 model. InD and OoD Dataset are TinyImageNet and CIFAR100 respectively. †, ‡, ♢ refer to OSR-based, SP-based, UOSR-based methods. Pre-training weights are used._
> |                  |        | UOSR    |        | OSR    |InC/InW |InC/OoD |InW/OoD |
> |:------------------|:--------:|:--------------:|:-------------:|:-------------:|:-------:|:-------------:|:-------------:|
> | Methods          | Acc. $\uparrow$  | AURC $\downarrow$        | AUROC $\uparrow$      | AUROC $\uparrow$      | AUROC $\uparrow$| AUROC $\uparrow$      | AUROC $\uparrow$      |
> | SoftMax †         | 77.02  | 340.70  | 86.22  | 77.27  | 88.83  | 85.62  | 49.27  |
> | ODIN †            | 77.23  | 359.57  | 84.01  | 75.32  | 86.53  | 83.43  | 47.81  |
> | LC †              | 77.23  | 385.15  | 79.76  | 71.85  | 83.21  | 78.98  | 47.69  |
> | OpenMax †         | 76.90  | 340.21  | 86.53  | 77.09  | 89.63  | 85.81  | 48.04  |
> | OLTR †            | 77.00  | 341.60  | 86.04  | 76.75  | 89.02  | 85.36  | 47.92  |
> | PROSER †          | 75.93  | 392.91  | 80.50  | 74.65  | 79.20  | 80.50  | 55.20  |
> | BCE ‡             | 76.91  | 339.43  | 86.40  | 76.83  | 89.64  | 85.66  | 47.44  |
> | TCP  ‡            | 77.82  | 336.61  | 86.48  | 77.47  | 89.72  | 85.76  | 48.38  |
> | DOCTOR  ‡         | 77.23  | 339.02  | 86.69  | 77.62  | 89.86  | 85.96  | 49.30  |
> | SIRC(MSP, z）♢     | 77.03  | 337.28  | 86.82  | 78.86  | 88.78  | 86.37  | 53.65  |
> | SIRC(MSP, Res.) ♢ | 77.03  | 316.14  | 90.66  | 87.00  | 88.82  | 91.08  | 73.31  |
> | SIRC(-H, z）♢      | 77.03  | 333.73  | 87.60  | 80.13  | 89.00  | 87.28  | 56.19  |
> | SIRC(-H, Res.) ♢  | 77.03  | 311.98  | 91.39  | 88.20  | 89.02  | 91.94  | 75.67  |
>
>
> ***
> Reference:
> [1] Sagar, et al. "Open-Set Recognition: A Good Closed-Set Classifier is All You Need" (ICLR 2022)
> [2] Guangyao, et al. "Adversarial Reciprocal Points Learning for Open Set Recognition" (TPAMI 2022)
> [3] Kong, et al. "OpenGAN: Open-Set Recognition via Open Data Generation" (ICCV 2021)
> [4] Lakshminarayanan, et al. "Simple and scalable predictive uncertainty estimation using deep ensembles" (NeurIPS 2017)
> [5] Hendrycks, et al. "Scaling Out-of-Distribution Detection for Real-World Settings" (ICML 2022)
> [6] Haoqi, et al. "ViM: Out-Of-Distribution with Virtual-logit Matching" (CVPR 2022)

---

> ### Author Response · Authors · 2022-11-18
> **Response to Reviewer zcng (3/4)**
>
> **[Q3]**: Evaluation results on SSB benchmark.
> **[A3]**: Thanks for the valuable suggestions to make our conclusion more convincing. We provide the UOSR results of two SSB datasets including CUB and FGCV-Aircraft[1] here in Table F and G for your reference. The resolution of ImageNet is too large for us which makes the batch size too small (224 * 224 compared to 32 * 32 in CIFAR100 or 64 * 64 in TinyImageNet). Instead, we provide the results when InD dataset is TinyImageNet and Ood dataset is CIFAR100 in Table H. TinyImageNet is more challenging and larger-scale than CIFAR100 when used as the training set.
>
> (1) From Table F and G we can see the UOSR and OSR performance are higher under the EASY mode compared to HARD mode as expected, since the OoD samples are more similar with InD samples in the HARD mode. Our conclusion that InW samples share similar uncertainty with OoD samples still holds, as the AUROC of InW/OoD is close to 50 and much lower than InC/OoD and InC/InW.
>
> (2) From Table H we can see that training the model on TinyImageNet is more challenging than CIFAR100, as the Acc. of CIFAR100 is 86.44 and Acc. of TinyImageNet only reaches 77.02. In this way, the impact of InW samples becomes further huge. For example, the performance gap between UOSR and OSR for the SoftMax method is 8.95 when InD dataset is TinyImageNet, and this value is only 0.34 when InD dataset is CIFAR100. So the InW samples are more important for the performance when InD dataset is challenging, as lower closed-set Acc. means more InW samples.
>
> **Table F** _UOSR benchmark in the image domain. All methods are conducted under the R50 model. Dataset is CUB-200-2011. †, ‡, ♢ refer to OSR-based, SP-based, UOSR-based methods. Pre-training weights are used. EASY/HARD modes are reported._
> |                  |        |     UOSR     |             | OSR         |InC/InW|InC/OoD      |InW/OoD      |
> |:------------------|:--------:|:--------------:|:-------------:|:-------------:|:-------:|:-------------:|:-------------:|
> | Methods          | Acc. $\uparrow$  | AURC $\downarrow$        | AUROC $\uparrow$      | AUROC $\uparrow$      | AUROC $\uparrow$| AUROC $\uparrow$      | AUROC $\uparrow$      |
> | SoftMax †          | 91.78  | 77.69/120.46 | 92.79/84.31 | 90.33/78.78 | 90.56 | 93.37/82.81 | 56.34/33.73 |
> | ODIN †             | 91.61  | 86.89/157.09 | 91.20/77.37 | 91.45/73.11 | 82.42 | 93.52/76.14 | 68.90/40.09 |
> | LC †               | 91.61  | 78.34/121.66 | 92.66/84.35 | 89.86/78.63 | 91.15 | 93.06/82.69 | 54.95/34.38 |
> | OpenMax †          | 91.30  | 78.07/119.61 | 92.87/85.43 | 90.01/79.78 | 91.14 | 93.35/83.98 | 54.99/35.67 |
> | OLTR †             | 91.33  | 80.38/118.50 | 92.43/85.30 | 89.42/79.72 | 90.66 | 92.91/83.95 | 52.65/35.19 |
> | PROSER †           | 91.33  | 79.39/128.14 | 92.50/83.81 | 90.32/78.53 | 89.31 | 93.37/82.42 | 58.21/37.60 |
> | BCE ‡             | 91.50  | 79.75/122.19 | 92.24/84.76 | 89.34/79.27 | 90.64 | 92.67/83.31 | 53.43/35.80 |
> | TCP ‡             | 92.06  | 77.31/116.93 | 92.55/84.85 | 90.28/79.92 | 90.07 | 93.17/83.64 | 56.71/36.78 |
> | DOCTOR ‡          | 91.61  | 78.09/121.61 | 92.76/84.37 | 90.09/78.68 | 91.13 | 93.18/82.71 | 56.26/34.70 |
> | SIRC(MSP, z）♢     | 91.78  | 78.04/119.42 | 92.72/84.46 | 90.19/78.95 | 90.59 | 93.27/83.00 | 55.78/33.70 |
> | SIRC(MSP, Res.) ♢  | 91.78  | 77.69/120.46 | 92.79/84.31 | 90.33/78.78 | 90.56 | 93.37/82.81 | 56.33/33.74 |
> | SIRC(-H, z）♢      | 91.78  | 76.97119.40  | 93.11/84.52 | 91.06/83.08 | 90.52 | 93.78/83.08 | 60.65/34.78 |
> | SIRC(-H, Res.) ♢   | 91.78  | 76.65/120.65 | 93.18/84.33 | 91.19/78.91 | 90.47 | 93.88/82.86 | 61.20/34.81 |

---

> ### Author Response · Authors · 2022-11-18
> **Response to Reviewer zcng (2/4)**
>
> **Table D** _UOSR benchmark in the image domain under the ResNet50 model. InD dataset is CIFAR100 while the OoD dataset is TinyImageNet. †, ‡, ♢ refer to OSR-based, SP-based, UOSR- based methods. OD: Outlier Data. N/G/R means No/Generated/Real OD. AUORC (%), AURC (×103) and Acc. (%) are reported. pt: pre-training._
> |                             |     |                 ||w/o pt||                  ||w/ pt||
> |:-----------------------------|:-----:|:--------:|:--------:|:--------:|:--------:|:--------:|:--------:|:--------:|:--------:|
> |                             |     |        |  UOSR   |         | OSR    |        |        UOSR     || OSR    |
> | Methods         | OD  |   ACC $\uparrow$  |  AURC $\downarrow$   | AUROC $\uparrow$  | AUROC $\uparrow$  | ACC $\uparrow$    | AURC $\downarrow$    | AUROC $\uparrow$  | AUROC $\uparrow$  |
> | ODIN†            | N   | 72.08  | 371.92  | 86.69  | 81.65  | 86.48  | 261.42  | 91.81  | 91.47  |
> | LC†              | N   | 72.08  | 372.55  | 84.16  | 76.37  | 86.48  | 280.69  | 87.68  | 85.16  |
> | OpenMax†         | N   | 73.64  | 361.22  | 85.33  | 76.23  | 86.43  | 261.53  | 90.29  | 85.86  |
> | MaxLogits†       | N   | 73.89  | 351.39  | 87.09  | 80.11  | 86.42  | 249.86  | 92.91  | 91.11  |
> | Entropy†         | N   | 73.89  | 355.54  | 86.25  | 78.14  | 86.43  | 257.36  | 91.21  | 87.22  |
> | OLTR†            | N   | 73.69  | 357.79  | 85.94  | 77.10  | 86.22  | 260.17  | 90.59  | 86.05  |
> | Ensemble†        | N   | 76.78  | 314.56  | 89.59  | 83.93  | 87.41  | 240.34  | 93.67  | 92.02  |
> | Vim†             | N   | 73.89  | 341.88  | 87.37  | 80.92  | 86.43  | 246.59  | 93.01  | 92.33  |
> | BCE‡             | N   | 73.29  | 369.91  | 84.29  | 74.59  | 86.39  | 257.74  | 90.79  | 86.50  |
> | TCP‡             | N   | 71.80  | 369.17  | 85.17  | 75.46  | 86.83  | 261.97  | 89.95  | 85.67  |
> | DOCTOR‡          | N   | 72.08  | 378.45  | 84.48  | 75.06  | 86.48  | 262.05  | 90.24  | 85.75  |
> | SIRC(MSP, z)♢    | N   | 73.13  | 358.93  | 85.77  | 76.67  | 86.44  | 260.08  | 90.53  | 85.98  |
> | SIRC(MSP, Res.）♢ | N   | 73.13  | 348.40  | 87.26  | 80.29  | 86.44  | 247.75  | 92.99  | 90.39  |
> | SIRC(-H, z）♢     | N   | 73.13  | 355.62  | 86.57  | 78.16  | 86.44  | 257.28  | 91.23  | 87.27  |
> | SIRC(-H, Res.）♢  | N   | 73.13  | 346.21  | 87.71  | 81.43  | 86.44  | 244.55  | 93.68  | 91.62  |
> | SoftMax †        | N   | 73.28  | 363.55  | 84.90  | 75.59  | 86.44  | 260.14  | 90.50  | 85.93  |
> | SoftMax (OE) †   | R   | 73.54  | 339.59  | 88.78  | 84.35  | 85.43  | 255.77  | 92.54  | 90.65  |
> | __ARPL †__           | __N__   | __73.03__  | __345.84__  | __88.13__  | __80.49__  | __84.67__  | __301.27__  | __87.01__  | __83.49__  |
> | __ARPL+CS †__        | __R__   | __72.78__  | __349.50__  | __87.65__  | __82.81__  | __83.60__  | __268.00__  | __92.00__  | __91.17__  |
> | MCD (Dropout ) †| N   | 76.49  | 375.01  | 82.25  | 79.21  | 87.21  | 301.71  | 83.57  | 81.69  |
> | MCD  †           | R   | 70.88  | 365.82  | 86.97  | 79.88  | 81.96  | 287.21  | 90.43  | 86.59  |
> | PROSER  †        | G   | 68.08  | 394.48  | 84.82  | 79.23  | 81.32  | 301.78  | 90.20  | 89.29  |
> | PROSER(EB) †     | R   | 71.82  | 366.65  | 86.06  | 81.95  | 85.06  | 269.79  | 90.84  | 90.38  |
> | VOS †            | G   | 73.44  | 356.68  | 86.65  | 79.72  | 86.62  | 249.24  | 92.94  | 91.37  |
> | VOS †            | R   | 73.18  | 331.12  | 89.96  | 85.78  | 85.93  | 251.44  | 92.92  | 91.34  |
> | __OpenGAN †__        | __R__   | __73.61__  | __334.04__  | __88.92__  | __85.48__  | __86.25__  | __260.19__  | __91.21__  | __90.94__  |
>
>
> **Table E** _Uncertainty distribution analysis in image domain with ResNet50. Pre-training is not used. InD dataset: CIFAR100. OoD dataset: TinyImageNet. AUORC (%) is reported. N/G/R means No/Generated/Real Outlier data._
> | Methods          | OD   | InC/OoD   | InC/InW  | InW/OoD  | OSR  | UOSR  |
> |:------------------|:--------:|:---------:|:--------:|:--------:|:--------:|:--------:|
> | ODIN |N |88.35 |80.76 |64.36 |81.65 |86.69
> | LC |N |84.60 |82.58 |55.14 |76.37 |84.16|
> | OpenMax |N |85.16 |85.96 |51.27 |76.23 |85.33
> | OLTR |N |85.99 |85.74 |52.22 |77.10 |85.94
> | SoftMax |N  |84.69 |85.68 |50.64 |75.59 |84.90
> |SoftMax(OE) |R  |90.04 |84.00 |68.54 |84.35 |88.78|
> |ARPL |N  |88.76 |85.77 |58.09 |80.49 |88.13|
> |ARPL+CS |R  |89.36 |81.40 |65.32 |82.81 |87.65|
> |MCD(Dropout) |N  |84.15 |74.17 |63.15 |79.21 |82.25|
> |MCD |R  |87.47 |85.23 |61.39 |79.88 |86.97|
> |PROSER |G |87.04 |77.84 |62.57 |79.23 |84.82|
> |PROSER (EB) |R  |88.36 |77.88 |65.60 |81.95 |86.06|
> |VOS |G |87.51 |83.38 |58.17 |79.72 |86.65 |
> |VOS |R  |91.44 |84.41 |70.32 |85.78 |89.96|
> ***

---

> ### Author Response · Authors · 2022-11-18
> **Response to Reviewer zcng (1/4)**
>
> We thank the reviewer for the critical comments that are very useful and helpful to improve our work. We are also happy to see that the reviewer acknowledges our finding that InW samples play a significant role in the performance. The answers of your concerns are provided below, and we recommend the reviewer to see other new insights and analysis in our modified manuscript, such as Appendix D (why InW samples have similar uncertainty with OoD samples from the feature and uncertainty estimation perspective) and other red paragraphs. The summary of the modification in our manuscript is in the 'Manuscript modification summary' public comment.
> ***
> **[Q1]**: It is straightforward and intuitive that InW samples have a strong impact on the final performance especially when the correlation between InD performance and OoD performance is founded [1].
> **[A1]**: (1) We agree that some researchers may infer from the conclusion in [1] that better closed-set performance brings better open-set performance is because of less InW samples, but __why less samples bring higher open-set performance is not discussed in [1] and not straightforward and intuitive.__ We first show that InW samples have similar uncertainty distributions with OoD samples rather than InC samples (AUROC of InW/OoD is close to 50 in Table 2 and 3 of the paper), which is contradictory with the ground truth of OSR. So InW samples are the primary false positive samples in the OSR problem, which is harmful for the OSR performance. The AUORC of InC/OoD is significantly higher than OSR further proves our claim (such as 84.69-75.59 in Table 4 of the paper). Therefore, fewer InW samples are beneficial for OSR performance. So why fewer InW samples bring higher OSR performance is not straightforward and intuitive and not discussed in [1]. The reviewer ZHBz also mentioned our strength that 'Although Vaze et al. [1] have demonstrated the importance of a good closed set, this paper further illustrates the impact of InW samples on OSR performance, providing a meaningful reference for future OOD detection research.'
>
> (2) __We think the conclusion in [1] does not weaken our additional value, but actually is a strong support of our finding.__ Like reviewer sZtz mentioned, there are no existing works that directly point out that the InW samples share similar uncertainty distribution with OoD samples rather than InC samples, including [1]. Therefore, our work and [1] support each other. On the one hand, our work provides a complementary explanation of the conclusion in [1], which can help more researchers understand why the conclusion in [1] happens. On the other hand, their conclusion is strong evidence of our conclusion. Besides, other analyses in our work including how pre-training, outlier data, and few-shot setting influence the behavior of InW samples cannot be inferred from [1] and are never discussed before. The reviewer q3SW also acknowledges that these analyses are exciting and vital.
>
> (3) __We believe that there are still many researchers who do not notice that the InW samples significantly impact the performance, as no OSR works consider the behavior of InW samples explicitly as far as we know.__ However, InW samples can influence the performance a lot (about 10 points AUORC as mentioned in (1)), so we hope our work can bring this finding to the OSR community and guide their work in the future.
> ***
> **[Q2]**: Why ARPL, ARPL+CS, and OpenGAN are missed in the comparison and analysis?
> **[A2]**: Thanks for recommending these well-known methods that help us build a more comprehensive benchmark. We include several additional methods into the UOSR benchmark, including ARPL[2], ARPL+CS[2], OpenGAN[3], Ensemble[4], MaxLogits[5], Entropy, and Vim[6]. The comprehensive results are in Table D. We can see that our finding that the UOSR performance is higher than OSR for the same method still holds, and the pre-training and outlier exposure are still beneficial for UOSR and OSR performance. One exception is ARPL has better UOSR performance than ARPL+CS w/o pre-training. The reason is that although ARPL+CS has better InC/OoD discrimination than ARPL (89.36-88.76 in Table E), the InC/InW discrimination is worse (81.40-85.77), which is consistent with our conclusion that outlier data is beneficial for InC/OoD performance but may deteriorate the InC/InW performance. So outlier data may not improve the overall UOSR performance (SoftMax has better UOSR performance than EB/VOS in the video domain (88.42-86.10/87.98 in Table 6 in the manuscript)). Please see the 'Analysis' Paragraph in Sec. 4 for more details.

---

### Official Review · Reviewer_q3SW · 2022-10-31

**Confidence:** 4
**Correctness:** 3
**Technical Novelty And Significance:** 2
**Empirical Novelty And Significance:** 2
**Recommendation:** 6

**Clarity, Quality, Novelty And Reproducibility:**

Regarding finding that entropy distributions are more closer to UOSR than OSR. We know entropy is a good anomlay detecion measure on-par with max-logit, max-softmax probability for OOD. For a well calibrated classifier we expect entropy to be high for both OOD & mis-classified samples. All visualizations of OSR/OOD in 2D shows OOD occupies center of viz or high entropy region especially for openset detection, meaning difficult to make decision which is the case for misclassified examples too.

Regarding devil is in wrongly classified samples, I think it is known phenomenon in openset detection works and [1] whole claim that good closed set improves open set detection also supports this. In addition, for e.g. in review of  Open-Set Recognition: A Good Closed-Set Classifier is All You Need, reviewer HAFU makes point about overlap of misclassified/incorrect predictions and openset predictions.


It is interesting to see that pre-training improves UOSR while outlier exposure does not, this can be expected as by pre-training we obtain richer or more-informative representations (lower bounded by training from scratch) whereas outlier exposure is more like a regularization technique w.r.t in-distribution decision boundaries & OE also makes the point that small amount of diverse data is useful further addition of data doesn't necessarily improve performance.

Regarding few shot UOSR, there are works on few shot open set detection but also GCD[4] and CCD[5], which do not even use labels as initial representation can cluster Near OOD classes reasonably well. [3] Decomposes representational capacity with unknown/anomaly detection measure like max-logit and argues that in-distribution classes-based representation has significant ability to detect OoD classes so not sure if few shot extension to UOSR is worthwhile novel contribution.

References:
[1] Vaze et al. Open-Set Recognition: A Good Closed-Set Classifier is All You Need (ICLR 2022)
[2] J. Kim et al. A Unified Benchmark for the Unknown Detection Capability of Deep Neural Networks
[3] R. Garrepalli et al. Oracle Analysis of Representations for Deep Open Set Detection
[4] Vaze et al. Generalized Category Discovery (CVPR 2022)
[5] Grow and Merge: A Unified Framework for Continuous Categories Discovery (NeurIPS 2022)

**Strength And Weaknesses:**

Strengths:

Paper is easy to follow and task of UOSR & evaluations is important to community.


Weakness:

Lacks novel insights, methods for UOSR based on observed phenomenon, please see next section for more details.
Is there any particular reason to not consider baselines like Deep ensembles [2] which is best performing method in initial UOSR evaluation benchmarking study? Also, as the paper makes a point w.r.t entropy distributions, it might be worthwhile to evaluate calibrated models (e.g. temperature scaling) and see the relative change in behavior compared to baseline models.

**Summary Of The Paper:**

In this work authors evaluate various open set recognition methods on the task of Unified Open set recognition(UOSR) where we need to detect novel classes but also misclassified/incorrect prediction. Authors also investigate training methods which reportedly improve open set recognition on UOSR. In addition, provide few-shot UOSR evaluation.


**Summary Of The Review:**

Authors evaluate methods on important task of UOSR but given lack of contributions in terms of providing new method, insight or unknown empirical behavior. Given UOSR is already established & evaluated in earlier work, and the emphasized phenomenon of wrongly classified examples well. Based on GCD, we know clustering works even in unsupervised fashion for OSR and hence few-shot learning would work well too albeit with lesser data. Although it is interesting to validate empirically that expectation of Uncertainty distributions is close to UOSR, in my opinion not sure if the work is worthwhile of ICLR acceptance.

---

> ### Author Response · Authors · 2022-11-18
> **Response to Reviewer q3SW (5/5)**
>
> **[Q6]**: I am not sure whether Few-shot UOSA is a novel contribution given GCD[12], CCD[13], and oracle analysis[14].
> **[A6]**: Thanks for your valuable comments. Although GCD[12] and CCD[13] can cluster unknown samples without labels, the training settings of few-shot UOSR and them differ a lot. (1) A significant reason that GCD[4] and CCD[5] can cluster the unknown classes well is that they provide the unknown samples during the training process, although labels of unknown classes are not given. So the model can gradually learn how to cluster unknown samples during training. In contrast, the model only sees the InD samples and cannot see unknown samples during training in our proposed few-shot UOSR. During evaluation, although we provide some reference OoD samples, the model cannot update itself based on the reference OoD samples. In summary, GCD and CCD can update the model based on the unknown samples while our few-shot UOSR cannot.
>
> (2) Our proposed few-shot UOSR is not contradictory with the Oracle analysis[14], as [14] shows that feature representations learned by InD classes have the significant ability for OoD detection, and few-shot UOSR also uses the features learned by the InD classes. OoD samples are not used in the training process, and they are only used in the evaluation process to provide reference OoD features. In addition, the introduced OoD samples during evaluation can further improve the UOSR performance given the features learned by InD samples (93.45/95.51 AUROC for UOSR and few-shot UOSR in the image domain; 93.38/96.19 AUROC for UOSR and few-shot UOSR in the video domain), which shows the significance of few-shot setting.
>
> (3) The practical application value of few-shot UOSR is promising. In many situations we may not have unlabeled unknown samples in advance so that we can use them during training like GCD and CCD. Usually we train a model based on the data we currently have, and after some time if we want to better recognize certain unknown samples during evaluation, we only need to provide 5 or 1 sample per unknown class and use them as reference samples to help the model identify these unknown samples. We do not have to update the model to fit the new unlabeled data, which is extremely convenient. In summary, the few-shot UOSR is practical and easy to implement in the real-world applications, and at the meantime brings higher performance, so we believe it is worth to be known by more researchers.
>
> ***
> Reference:
> [1] Lakshminarayanan, et al. "Simple and scalable predictive uncertainty estimation using deep ensembles" (NeurIPS 2017)
> [2] Hendrycks, et al. "Scaling Out-of-Distribution Detection for Real-World Settings" (ICML 2022)
> [3] Haoqi, et al. "ViM: Out-Of-Distribution with Virtual-logit Matching" (CVPR 2022)
> [4] Guangyao, et al. "Adversarial Reciprocal Points Learning for Open Set Recognition" (TPAMI 2022)
> [5] Kong, et al. "OpenGAN: Open-Set Recognition via Open Data Generation" (ICCV 2021)
> [6] Chuan, et al. "On calibration of modern neural networks." (ICML 2017)
> [7] Hendrycks, et al. "Using Pre-Training Can Improve Model Robustness and Uncertainty" (ICML 2019)
> [8] Kaiming, et al. "Rethinking ImageNet Pre-training" (ICCV 2019)
> [9] Sagar, et al. "Open-Set Recognition: A Good Closed-Set Classifier is All You Need" (ICLR 2022)
> [10] Jihyo, et al. "A Unified Benchmark for the Unknown Detection Capability of Deep Neural Networks" (arXiv)
> [11] Guoxuan, et al. "Augmenting softmax information for selective classi- fication with out-of-distribution data" (arXiv)
> [12] Sagar, et al. "Generalized Category Discovery" (CVPR 2022)
> [13] Xinwei, et al. "Grow and Merge: A Unified Framework for Continuous Categories Discovery" (NeurIPS 2022)
> [14] Risheek, et al. "Oracle Analysis of Representations for Deep Open Set Detection" (arXiv)

---

> ### Author Response · Authors · 2022-11-18
> **Response to Reviewer q3SW (4/5)**
>
> **[Q4]**: Lacks novel methods for the observed phenomenon of UOSR.
> **[A4]**: (1) We totally agree that novel method proposing is one of the effective ways to push the development of one research direction, but some important findings and analysis are also indispensable to pave the right way for the following researchers to design their new methods, such as [7,8]. Our paper belongs to the latter kind of  work, which aims to deeply analyze the UOSR problem and point out InW samples actually share the similar uncertainty with OoD samples rather than InC samples. On the one hand, __we provide a new and more detailed perspective to analyze the existing OSR method.__ For example, LC improves the OSR performance not because it has better InC/OoD performance compared to the Softmax baseline (LC: 84.60, SoftMax: 84.69), but because of lower InC/InW discrimination (LC: 82.58, SoftMax: 85.68) which is actually not related to OoD at all. On the other hand, __our findings can inspire the following researchers to take our findings into account when they design new methods.__ For example, we should improve InC/InW discrimination for better UOSR performance but eliminate InC/InW discrimination for better OSR performance, or whether we should use outlier data or pre-training weights or few-shot settings when we design a new method. So although the main part of this work is not to design a single new method, we believe our findings and analysis can guide the future researchers to effectively design more and more methods.
>
> (2) Based on our analysis in (1) that both better InC/InW and InC/OoD discrimination are important for the UOSR problem, we further propose the simple but effective FS-KNNS in few-shot UOSR and achieves state-of-the-art performance. We show how to improve the InC/OoD discrimination without losing InC/InW discrimination for better UOSR performance in the few-shot setting. Our FS-KNNS has the best overall UOSR performance and much better InC/InW performance than baselines (FS-KNN/SSD: 79.58/70.14; FS-KNNS: 87.98). Our designing process is an effective and straightforward example for future researchers of how to take InW samples behavior into account when proposing new methods.
> ***
> **[Q5]**: Entropy distributions of InW samples are closer to OoD samples is well-known and supported by [9] and its reviewer.
> **[A5]**: We are very happy to see that some researchers may also notice this phenomenon, which strengthens that our work is meaningful. We believe this conclusion is not well-known enough yet, as no existing paper has pointed out this phenomenon directly (just as Reviewer sZtz said), including the proposer of UOSR [10,11] and [9].
>
> (1) Although some researchers may notice this phenomenon, __their understanding of this phenomenon may be limited and incomplete.__ Instead, we conduct comprehensive experiments to testify that it is a common and general phenomenon and ensure that this conclusion is correct. In this way, some researchers who may accidentally find this phenomenon but are not very sure can use this conclusion without worry. In addition, how outlier data and pre-training influence the InW behavior is even harder to be noticed and we reveal the corresponding results to more researchers.
>
> (2) __We believe that there are still lots of researchers who may not notice this significant phenomenon, as no OSR works consider the behavior of InW samples explicitly as far as we know.__ However, this phenomenon can influence the performance a lot (without InW samples, the OSR performance significantly improves from 75.59 to 84.69 of SoftMax method based on Table 4 of the manuscript), so we hope our work can bring this conclusion to the OSR community and guide their work in the future.
>
> (3) __The proposer of UOSR [10,11] does not explicitly point out that the uncertainty distribution of InW samples is more similar with OoD samples instead of InC samples,__ although they illustrated that rejecting InW samples is practical. In contrast, we first find that the AUORC of UOSR is higher than OSR for the same method, and then we show the reason is that the InW samples have similar uncertainty scores with OoD samples. So our work provides a deeper analysis of [10,11]. As for [9], they concluded better closed-set performance means better open-set performance, __but they do not explicitly point out a significant reason of this conclusion is that InW samples become less with higher closed-set accuracy.__ In contrast, we find the InW samples have similar uncertainty with OoD samples rather than InC samples, so InW samples can deteriorate the OSR performance, as mentioned in (2) and illustrated in the last paragraph 'A better closed-set model is better for UOSR' in Sec. 3. This is one important reason for the conclusion in [9]. So our finding is a complementary explanation of [9], which helps more researchers deeply understand why the conclusion of [9] happens, and their [9] conclusion is strong support of our finding.

---

> ### Author Response · Authors · 2022-11-18
> **Response to Reviewer q3SW (3-2/5)**
>
> Let us give a toy example of how $f$ changes the similarity relationship. Suppose the logit space is under 2 dimensions $(s,t)$, and $x_c=(2,2), x_w=(1,1), x_o=(1,3)$ are the logits of an InC, InW, and OoD sample. In this case, $x_w$ is closer to $x_c$ than $x_o$ (smaller L2 distance), which means the InW sample has a similar feature to the InC sample. If $f(x)=-s$ then $u_c=-2, u_w=-1, u_o=-1$. In this case, $ u_w=u_o>u_c$. So the uncertainty score of the InW sample is similar to the OoD sample instead of InC sample. This example illustrates why an InW sample has a similar feature to an InC sample, but has a similar uncertainty score with OoD sample. This is how existing uncertainty estimation methods work, as the results in Sec. 3 and Sec. 4 show that InW samples have similar uncertainty distribution with OoD samples. This kind of method is suitable for UOSR problem where InW and OoD samples are supposed to be rejected at the same time. Please see Fig. 12 (a) in Appendix D for visualization.
>
> We provide another uncertainty estimation function $f$ in Fig. 13 (b) in Appendix D, where $f(x)=t/s$. In this case, $ u_w=u_c=1<u_o=3$, so the uncertainty score of the InW sample is similar to the InC sample instead of the OoD sample. This is the ideal case for the traditional OSR problem to reject OoD samples and accept InC and InW samples. Although existing uncertainty estimation methods are designed to behave like Fig. 12 (b) for the OSR problem, they actually work as Fig. 12 (a) which is more suitable for the UOSR problem.

---

> ### Author Response · Authors · 2022-11-18
> **Response to Reviewer q3SW (3-1/5)**
>
> **[Q3]**: Lacks novel insights for the observed phenomenon of UOSR.
> **[A3]**: Your concern encourages us to think deeper into our finding that InW samples have more similar uncertainty scores with OoD samples rather than InC samples. We begin with considering whether this phenomenon comes from the feature distribution perspective, i.e., the InW samples have close features with OoD samples and are far away from InC samples. We find that the features of InC, InW, and OoD samples are actually separable and surprisingly find that InW samples have more similar features to InC samples than OoD samples, which may seem contradictory with our finding that InW samples share more similar uncertainty scores with OoD samples than InC samples. To better help readers understand why these two contradictory phenomenons happen, we provide a toy example to show how uncertainty estimation methods influence the similarity relationship in Appendix D.
>
> (1) __Features of InC/InW/OoD samples are separable.__ We find the feature distribution of InC, InW and OoD samples follow a hierarchy structure. The InW samples surround the InC samples, and OoD samples are further far away and located at the outer edge of InW samples. Please see Fig. 10 in Appendix D for t-SNE visualization results. To further testify this idea, we calculate the similarity between InC/InW/OoD samples and training samples of each class, as shown in Fig. 11 in Appendix D. We can see that the InC samples are the most similar samples with training samples, and InW samples have smaller similarity and OoD samples have the smallest similarity. Therefore, features of InC/InW/OoD samples are in a hierarchy structure and distinguishable.
>
> (2) __Features of InW samples are more similar to InC samples than OoD samples.__ As (1) shows that InC/InW/OoD samples do not overlap with each other in the feature space, we calculate the similarity between InW/InC samples and InW/OoD samples in the feature space and logit space. Then we provide the mean of uncertainty scores based on the KNN method and MaxLogit method, as well as the AUROC of InW/InC and InW/OoD to illustrate the uncertainty discrimination performance. In Table C we can see that the feature similarity of InW/InC is higher than InW/OoD (0.797-0.733), which means InW samples have more similar features with InC samples than OoD samples. However, the uncertainty scores of InW samples are more similar with OoD samples than InC samples (0.158/0.179-0.057), so that InW/OoD can not be distinguished very well like InW/InC (57.25-84.91). Therefore, the feature similarity relationship is contradictory with the uncertainty score similarity relationship, as InW samples have similar features with InC samples but have similar uncertainty scores with OoD samples. So the reason that InW samples share similar uncertainty distribution with OoD samples is not they have similar features.
>
> **Table C** _We provide the feature similarity of InW/InC and InW/OoD, the mean of uncertainty score, and the AUROC of InW/InC and InW/OoD in this table._
> |               |Similarity|         ||Uncertainty   |  |Discrimination (AUROC)  ||
> |:---------------|:----------:|:---------:|:-------:|:-------:|:-------:|:---------:|:---------:|
> |               | InW/InC  | InW/OoD | Inc   | InW   | OoD   | InW/InC | InW/OoD |
> | Feature space | 0.797    | 0.733   | 0.057 | 0.158 | 0.179 | 84.91   | 57.25   |
> | Logit space   | 0.718    | 0.571   | 0.314 | 0.537 | 0.583 | 83.36   | 57.95   |
>
> (3) __Uncertainty estimation methods are the reason for InW and OoD samples have similar uncertainty scores.__ The uncertainty score is defined as $u=f(x)$, where $x$ refers to the feature representation and $f$ refers to the uncertainty estimation function. Since the reason that InW samples share similar uncertainty distribution with OoD samples does not lie in the feature perspective according to (2), the only reason is in the uncertainty estimation methods.

---

> ### Author Response · Authors · 2022-11-18
> **Response to Reviewer q3SW (2/5)**
>
> **[Q2]**: Evaluation results of temperature scaling.
> **[A2]**: Thanks for the very nice suggestion. In the manuscript of the old version, we discuss the performance relationship between UOSR and model calibration (MC) in Appendix A, and show that they are not perfectly positively correlated. Now with your valuable suggestion, we further use the temperature scaling method to support our original claim. We show the results of temperature scaling in Table B here and also in Appendix B of the modified manuscript. The model calibration performance is measured with ECE[6]. We can see that the optimal temperature $T$ for MC ($T=2$) is not the best case for UOSR. When $T$ grows, the InC/OoD discrimination increases, but the InC/InW discrimination drops. Therefore, the OSR performance keeps improving with larger $T$, but the UOSR may not benefit from larger $T$ because of lower InC/InW discrimination. For example, the best $T$ for UOSR with pre-training is 5 rather than 20. But in general, temperature scaling is a simple and useful technique for both MC and UOSR, as $T=2$ has better MC and UOSR performance than $T=1$ (without temperature scaling). The only drawback of temperature scaling is it needs the validation set to determine the optimal $T$. We also show how temperatures influence the uncertainty distribution in Appendix B. Feel free to watch them as we cannot put figures here.
>
> **Table B** _UOSR and MC performance under different temperatures $T$. pt: pre-training._
> |     |       |         |w/o pt||        |         |        |         |w/   pt||        |         |
> |:-----|:-------:|:---------:|:-------:|:--------:|:---------:|:---------:|:--------:|:---------:|:--------:|:--------:|:---------:|:---------:|
> |     | ECE $\downarrow$  | AURC $\downarrow$   |       |  AUROC $\uparrow$ |  |         | ECE $\downarrow$   | AURC $\downarrow$   |        |   AUROC $\uparrow$| |         |
> | T   | MC    | UOSR    | UOSR  | OSR    | InC/InW | InC/OoD | MC     | UOSR    | UOSR   | OSR    | InC/InW | InC/OoD |
> | 0.1 | 0.247 | 355.69  | 66.58 | 62.06  | 66.96   | 66.48   | 0.128  | 257.20  | 69.07  | 66.67  | 68.49   | 69.15   |
> | 0.5 | 0.207 | 351.39  | 84.10  | 74.87  | 85.01   | 83.86   | 0.106  | 257.26  | 88.66  | 83.71  | 88.61   | 88.67   |
> | 1   | 0.146 | 358.31  | 85.57 | 76.90  | __85.18__   | 85.67   | 0.081  | 260.14  | 90.51  | 85.93  | __89.58__   | 90.64   |
> | 2   | __0.119__ | 352.60  | 86.79 | 79.19  | 84.62   | 87.35   | __0.018__  | 250.99  | 92.40  | 89.09  | 89.08   | 92.85   |
> | 5   | 0.344 | 351.45  | 87.05 | 79.94  | 84.05   | 87.84   | 0.523  | __249.00__  | __92.96__  | 90.86  | 86.00   | 93.90   |
> | 10  | 0.256 | 351.39  | 87.08 | 80.04  | 83.94   | 87.90   | 0.509  | 249.55  | 92.93  | 91.02  | 85.22   | 93.97   |
> | 20  | 0.197 | __351.38__  | __87.09__ | __80.08__  | 83.89   | __87.92__   | 0.414  | 249.73  | 92.92  | __91.07__  | 84.99   | __93.99__   |

---

> ### Author Response · Authors · 2022-11-18
> **Response to Reviewer q3SW (1/5)**
>
> We thank the reviewer q3SW for the careful comments and constructive suggestions, which effectively help us enrich our work and encourage us to think deeper about the reason of our findings. We have some new insights about the conclusion in our work, please see the answers below to see the improvements based on your valuable feedback. The summary of the modification in our manuscript can be found in the 'Manuscript modification summary' public comment.
>
> **[Q1]**: Evaluation results of deep ensembles.
> **[A1]**: Thanks for the general reminder. To address your comments, we include several additional methods into the UOSR benchmark, including Ensemble[1], MaxLogits[2], Entropy, Vim[3], ARPL[4], ARPL+CS[4] and OpenGAN[5]. The comprehensive results are in Table A. We can see the deep ensemble achieves the best performance as expected, and our finding that the UOSR performance is better than OSR for the same method still holds for the deep ensemble method (89.59/83.93 w/o pre-training and 93.67/92.02 w/ pre-training), which indicates InW samples have similar uncertainty scores with OoD samples rather than InC samples.
>
> **Table A** _UOSR benchmark in the image domain under the ResNet50 model. InD dataset is CIFAR100 while the OoD dataset is TinyImageNet. †, ‡, ♢ refer to OSR-based, SP-based, UOSR- based methods. OD: Outlier Data. N/G/R means No/Generated/Real OD. AUORC (%), AURC (×103) and Acc. (%) are reported. pt: pre-training._
> |                             |     |                 ||w/o pt||                  ||w/ pt||
> |:-----------------------------|:-----:|:--------:|:--------:|:--------:|:--------:|:--------:|:--------:|:--------:|:--------:|
> |                             |     |        |  UOSR   |         | OSR    |        |        UOSR     || OSR    |
> | Methods†         | OD  |   ACC $\uparrow$  |  AURC $\downarrow$   | AUROC $\uparrow$  | AUROC $\uparrow$  | ACC $\uparrow$    | AURC $\downarrow$    | AUROC $\uparrow$  | AUROC $\uparrow$  |
> | ODIN†            | N   | 72.08  | 371.92  | 86.69  | 81.65  | 86.48  | 261.42  | 91.81  | 91.47  |
> | LC†              | N   | 72.08  | 372.55  | 84.16  | 76.37  | 86.48  | 280.69  | 87.68  | 85.16  |
> | OpenMax†         | N   | 73.64  | 361.22  | 85.33  | 76.23  | 86.43  | 261.53  | 90.29  | 85.86  |
> | MaxLogits†       | N   | 73.89  | 351.39  | 87.09  | 80.11  | 86.42  | 249.86  | 92.91  | 91.11  |
> | Entropy†         | N   | 73.89  | 355.54  | 86.25  | 78.14  | 86.43  | 257.36  | 91.21  | 87.22  |
> | OLTR†            | N   | 73.69  | 357.79  | 85.94  | 77.10  | 86.22  | 260.17  | 90.59  | 86.05  |
> | __Ensemble†__        | __N__   | __76.78__  | __314.56__  | __89.59__  | __83.93__  | __87.41__  | __240.34__  | __93.67__  | __92.02__  |
> | Vim†             | N   | 73.89  | 341.88  | 87.37  | 80.92  | 86.43  | 246.59  | 93.01  | 92.33  |
> | BCE‡             | N   | 73.29  | 369.91  | 84.29  | 74.59  | 86.39  | 257.74  | 90.79  | 86.50  |
> | TCP‡             | N   | 71.80  | 369.17  | 85.17  | 75.46  | 86.83  | 261.97  | 89.95  | 85.67  |
> | DOCTOR‡          | N   | 72.08  | 378.45  | 84.48  | 75.06  | 86.48  | 262.05  | 90.24  | 85.75  |
> | SIRC(MSP, z)♢    | N   | 73.13  | 358.93  | 85.77  | 76.67  | 86.44  | 260.08  | 90.53  | 85.98  |
> | SIRC(MSP, Res.）♢ | N   | 73.13  | 348.40  | 87.26  | 80.29  | 86.44  | 247.75  | 92.99  | 90.39  |
> | SIRC(-H, z）♢     | N   | 73.13  | 355.62  | 86.57  | 78.16  | 86.44  | 257.28  | 91.23  | 87.27  |
> | SIRC(-H, Res.）♢  | N   | 73.13  | 346.21  | 87.71  | 81.43  | 86.44  | 244.55  | 93.68  | 91.62  |
> | SoftMax †        | N   | 73.28  | 363.55  | 84.90  | 75.59  | 86.44  | 260.14  | 90.50  | 85.93  |
> | SoftMax (OE) †   | R   | 73.54  | 339.59  | 88.78  | 84.35  | 85.43  | 255.77  | 92.54  | 90.65  |
> | ARPL †           | N   | 73.03  | 345.84  | 88.13  | 80.49  | 84.67  | 301.27  | 87.01  | 83.49  |
> | ARPL+CS †        | R   | 72.78  | 349.50  | 87.65  | 82.81  | 83.60  | 268.00  | 92.00  | 91.17  |
> | MCD (Dropout ) †| N   | 76.49  | 375.01  | 82.25  | 79.21  | 87.21  | 301.71  | 83.57  | 81.69  |
> | MCD  †           | R   | 70.88  | 365.82  | 86.97  | 79.88  | 81.96  | 287.21  | 90.43  | 86.59  |
> | PROSER  †        | G   | 68.08  | 394.48  | 84.82  | 79.23  | 81.32  | 301.78  | 90.20  | 89.29  |
> | PROSER(EB) †     | R   | 71.82  | 366.65  | 86.06  | 81.95  | 85.06  | 269.79  | 90.84  | 90.38  |
> | VOS †            | G   | 73.44  | 356.68  | 86.65  | 79.72  | 86.62  | 249.24  | 92.94  | 91.37  |
> | VOS †            | R   | 73.18  | 331.12  | 89.96  | 85.78  | 85.93  | 251.44  | 92.92  | 91.34  |
> | OpenGAN †        | R   | 73.61  | 334.04  | 88.92  | 85.48  | 86.25  | 260.19  | 91.21  | 90.94  |

---

> ### Comment · Reviewer_q3SW · 2022-12-12
> **Increasing Rating based on author feedback**
>
> First would like to thank authors for addressing many of raised issues in constructive way!
>
> It is interesting to see that as temperature increase OSR Performance improves but not UOSR, as this is just a scalar manipulation and it might point to relative similarity of Incorrect samples & OoD to correctly classified samples. Not sure why this is, is it because of batchnorm or some other characteristics of representation/deep net especially when comparing hard OSR samples to detect & wrongly classified samples as both type of samples should be near decision boundary.
>
> Authors also present experiments in terms of similarity & uncertainty measure, in some ways we can consider these being different measurement signals w.r.t OSR/UOSR/OoD detection and based on scoring method what changes is relative importance of each feature, etc. e.g. max logit weighs penultimate layer  by linear weights where as knn/cosine distance has equal importance to each feature.
>
> But the fact that InC/OoD are more difficult to separate is still consistent with overall hypothesis that InW might be more similar to OoD.
>
> As authors comment on InW samples closer to InC than OoD, if possible would encourage authors to separate 'hard and easy' samples of OoD & their distribution in comparison to InC,InW, could be based on their entropy or any uncertainty/confidence measure. Based on current results it would be interesting to see where hard OoD distributions lie w.r.t InC & InW samples.

---

### Author Response · Authors · 2022-11-18
**Manuscript Modification Summary**

We thank all reviewers for their detailed and constructive suggestions. Based on these careful comments, we enrich our manuscript a lot and the modification parts (red sentences in the new manuscript) are summarized below:
***
1. Second paragraph of Sec. 1. We reorganize our main finding.

2. 'Anomaly/Outlier Detection (AD/OD)' paragraph of Sec. 2. We add the related work of outlier detection.

3. 'Importance' paragraph of Sec. 3. We illustrate the importance of our finding to the community.

4. 'Why' paragraph of Sec. 3. We analyze the underlying reason for our finding from the feature distribution perspective.

5. 'Results' paragraph and Table 5 of Sec. 4. We add more methods into the UOSR benchmark and reorganize the Table 5. Fair comparison w.r.t. outlier data and discussion of generated or real outlier are included.

6. Appendix B. We evaluate the temperature scaling method on the model calibration and UOSR problems.

7. Appendix D. We find InW samples have more similar features to InC samples rather OoD samples, and explain why it is contradictory with that InW have more similar uncertainty scores with OoD samples.

8. Appendix E. We add the definition of AURC.

9. Appendix G. We provide the UOSR results under the traditional OSR dataset settings, where a part of data within one dataset is regarded as InD and the remaining data is regarded as OoD.

10. Appendix H. We provide the UOSR results when InD dataset is TinyImageNet and OoD dataset is CIFAR100.

11. Appendix I. We provide the UOSR results on SSB datasets to show that our conclusion still holds for fain-grained datasets.

12. Appendix K. We provide the UOSR results when noisy outlier data is used, and use NGC to clean corrupted outlier data.

---

### Decision · Program_Chairs · 2023-01-20

**Decision:**

Accept: poster

**Justification For Why Not Higher Score:**

Interesting paper, but novelty is somewhat limited and problem domain is fairly narrow.

**Justification For Why Not Lower Score:**

It is a good paper that was executed well.

**Metareview: Summary, Strengths And Weaknesses:**

Paper was reviewed by four reviewers. Post-rebuttal it has received the following ratings:

* 2 x marginally above the acceptance threshold
* 1 x marginally below the acceptance threshold
* 1 x accept, good paper

Reviewers generally agree that the paper is well-written, easy to follow, and addresses an important problem. They also point to some shortcoming. Mainly, while paper conducts a through analysis of the UOSR problem, gained empirical insights are limited [q3SW] and novelty is somewhat limited as well [zcng]. On the other hand, [sZtz] is much more positive and argues that paper makes "important point which has not received sufficient emphasis in the literature". Overall, 3 out of 4 reviewers argue for a form of acceptance.

AC has read the reviews, rebuttal and discussion that followed. AC agrees with [sZtz] and others that while some aspects of findings may have been hinted at in other works, the proposed paper still makes interesting and important empirical observations about an significant problem / setting. It is also well executed in terms of exposition, through experimentation and discussions. As such, in the opinion of the AC, the paper would make a valuable contribution to the ICLR conference.

**Note From Pc:**

if the above contains the word "oral" or "spotlight" please see: "oral" presentation means -> notable-top-5% and "spotlight" means -> notable-top-25%. As stated in our emails, we are disassociating presentation type from AC recommendations